# Constrained Reinforcement Learning as Wasserstein Variational Inference: Formal Methods for Interpretability

## Abstract

Reinforcement learning can provide effective reasoning for sequential decision-making problems with variable dynamics. Such reasoning in practical implementation, however, poses a persistent challenge in interpreting the reward function and corresponding optimal policy. Consequently, representing sequential decision-making problems as probabilistic inference can have considerable value, as, in principle, the inference offers diverse and powerful mathematical tools to infer the stochastic dynamics whilst suggesting a probabilistic interpretation of policy optimization. In this study, we propose a novel Adaptive Wasserstein Variational Optimization, namely AWaVO, to tackle these interpretability challenges. Our approach uses formal methods to achieve the interpretability of guaranteed convergence, training transparency, and sequential decisions. To demonstrate its practicality, we showcase guaranteed interpretability including a global convergence rate $\Theta(1/\sqrt{T})$ not only in simulation but also in real-world robotic tasks. In comparison with state-of-the-art benchmarks including TRPO-IPO, PCPO and CRPO, we empirically verify that AWaVO offers a reasonable trade-off between high performance and sufficient interpretability. The real-world hardware implementation is demonstrated via an anonymous video [1].

## 1 Introduction

The representation of sequential decision-making problems as Reinforcement Learning (RL) or optimal control provides an effective means of reasoning optimal policies or control strategies in the presence of uncertainties Levine (2018). Nevertheless, such reasoning encounters a persistent difficulty to create a convincing interpretation of the sequential decision-making and its corresponding optimal policies Devidze et al. (2021); Levine (2022). This challenge in comprehension poses a significant barrier to the real-world implementation and adoption of RL in domains like advanced manufacturing Napoleone et al. (2020), autonomous systems Fernandez-Llorca & Gómez (2023), healthcare Albahri et al. (2023), and financial trading McNamara (2016).

**Key Challenges.** The complexity of interpretability within the context of RL can be conceptualized through three distinct phases: **a. Guarantee of convergence** ensures that a RL framework converges towards an optimal policy, e.g., in an asymptotic manner. **b. Transparency in convergence (or training)** focuses on discerning the underlying mechanism through which a RL algorithm achieves convergence towards an optimal or nearly optimal policy. An instance is the convergence rate, where, based on a given number of training iterations, the rate enables the prediction of the expected level of convergence with a certain degree of confidence. **c. Interpretation of decisions** involves clarifying the extent to which latent factors influence the sequential decisions made, which is an aspect of paramount importance particularly in instances where decisions lead to unfavorable outcomes. Moreover, due to legal mandates in industries, such as ensuring the trustworthiness of self-driving vehicles Fernández Llorca & Gómez (2021); Fernandez-Llorca & Gómez (2023), aerospace engineering Brat (2021); Torens et al. (2022), and high-frequency trading McNamara (2016), this facet of interpretation is of even greater significance.

---

[1] https://youtu.be/LZJs1U778XU

One widely adopted approach to achieving model interpretability involves the use of post-hoc explanation methods. These methods provide retrospective rationales for model predictions, often through the creation of saliency maps or exemplars, as discussed in previous research Lipton (2018); Kenny et al. (2021). These approaches, while popular, can sometimes yield incomplete or inaccurate explanations Slack et al. (2020). In response to these limitations, recent research has shifted focus towards intrinsic interpretability, as detailed in prior studies Rudin (2019); Kenny et al. (2022). The fundamental idea behind this shift is to develop inherently interpretable models, allowing for a transparent and understandable view of the decision-making process. This transparency enables the calibration of user trust and facilitates the prediction of the system's capabilities, as outlined in **Key Challenges**.

To our best knowledge, we present the first intrinsically interpretable constrained RL framework by addressing sequential decision-making problems through the lens of probabilistic inference. Specifically, we reframe constrained RL as Wasserstein variational optimization, leveraging an enhanced foundational inference framework known as augmented Probabilistic Graphical Models (PGMs), as illustrated in Figure 1 (Section 3). Our proposed **A**daptive Sliced **Wa**sserstein **V**ariational **O**ptimization (AWaVO), as elaborated in Figure 2 (Section 4), consists of two primary steps: a. *Policy Updating:* Primal Policy Optimization using Distributional Representation (PPO-DR) is conducted to address dynamic uncertainties adaptively, as shown in Algorithm 1 (Section 4.2). More importantly, PPO-DR enhances the transparency of the convergence process, thereby addressing a significant portion of the deficiencies outlined in **Key Challenges a. and b.**; b. *Inference Execution:* Wasserstein Variational Inference (WVI), as detailed in Section 4.1, is subsequently performed to achieve the probabilistic interpretation of decisions, thereby tackling **Key Challenges c.**

Our contributions can be summarized: 1) **Adaptive Generalized Sliced Wasserstein Distance**, referred to as A-GSWD, incorporates the Sliced Wasserstein Distance (SWD) along with adaptive Radon transforms and a variational distribution. To handle dynamic uncertainties, the proposed A-GSWD adaptively determines the slicing directions of hypersurfaces to enhance the precision of distribution distance computation; 2) **Adaptive Sliced Wasserstein Variational Optimization**, abbreviated as AWaVO, employs inference to reformulate the problem of sequential decision-making. To tackle the **Key Challenges**, AWaVO leverages PPO-DR to enhance the transparency of convergence under dynamic uncertainties. Additionally, WVI is employed to provide a probabilistic interpretation of decisions; 3) **Formal methods for interpretation** are employed to demonstrate theoretical comprehension on metric judgment of A-GSWD, transparency of training convergence, and probabilistic interpretation of sequential decisions.

## 2 RELATED WORK

**Reinforcement Learning as Inference.** The relationship between sequential decision-making and probabilistic inference has been explored extensively in recent years Levine (2018); Okada & Taniguchi (2020); Liu et al. (2022). Despite variations in terminology, the core inference frameworks remains consistent, namely, PGMs Koller & Friedman (2009). While substantial research exists on learning and inference techniques within PGMs Levine (2018), the direct connection between RL (or control) and probabilistic inference is not immediately apparent. Welch et al. (1995) establishes that control and inference are dual perspectives of the same problem. This connection offers novel insights and enhanced understanding within control problems by leveraging mathematical tools of inference Toussaint & Storkey (2006); Kappen et al. (2012). Moreover, the study on 'RL as inference' represents another prominent trend. Specifically, Levine (2018) demonstrates that RL is equivalent to probabilistic inference under dynamics. Chua et al. (2018); Okada & Taniguchi (2020) approach dynamics modeling by employing Bayesian inference optimization. Furthermore, O'Donoghue et al. (2020) revisits the formalization of 'RL as inference' and demonstrates that with a slight algorithmic modification, this approximation can perform well even in problems where it initially performs poorly. In this study, we formalize constrained RL as Wasserstein variational optimization to achieve decision-interpretations.

**Optimal Transport Theory.** Forming effective metrics between two probability measures is a fundamental challenge in machine learning and statistics communities. The optimal transport theory, particularly the Wasserstein distance, has garnered significant attention across various domains Solomon et al. (2014); Kolouri et al. (2017); Schmitz et al. (2018); Wang & Boyle (2023) due to its accuracy, robustness, and stable optimization. Nevertheless, it can be computationally demanding, especially with high-dimensional data. Recent advancements emphasize computational efficiency

through differentiable optimization Peyré et al. (2017). Among these methods, Sinkhorn distance Cuturi (2013); Altschuler et al. (2017) introduces entropy regularization to smoothen the convex regularization. Another notable approach involves slicing or linear projection Ng (2005), i.e., Sliced Wasserstein Distance (SWD) Bonneel et al. (2015), which leverages the measures' Radon transform for efficient dimensionality reduction. Then, variants of SWD such as Generalized SWD (GSWD) Kolouri et al. (2019) improves projection efficiency. These advancements contribute to the efficiency in optimal-transport-based metrics. However, they suffer from reduced accuracy as SWD only slices distributions using linear hyperplanes, which may fail to capture the complex structures of data distributions. To overcome the accuracy limitation, Augmented SWD (ASWD) Chen et al. (2020) projects onto flexible nonlinear hypersurfaces, enabling the capture of intricate data distribution structures. Building upon the ASWD framework, we introduce an adaptive variant called A-GSWD which leverages the projection onto nonlinear hypersurfaces and combines it with PPO-DR to achieve adaptivity. This adaptive approach enhances the efficiency and accuracy of Wasserstein distance computation, improving upon the limitations of previous methods.

## 3 SEQUENTIAL DECISION-MAKING AS PROBABILISTIC INFERENCE

**Sequential Decision-making Problems as Inference.** A sequential decision-making problem, formalized as a standard RL or control problem, can be seen as an inference problem Levine (2018):

$$p(\tau|\mathcal{O}_{0:T-1} = 1) \propto \int \underbrace{\prod_{t=0}^{T-1} p(\mathcal{O}_t = 1|\boldsymbol{s}_t, \boldsymbol{a}_t)}_{:=p(\mathcal{O}|\tau)} \cdot p(\boldsymbol{s}_0) \underbrace{\left\{ \prod_{t=0}^{T-1} p(\boldsymbol{a}_t|\boldsymbol{s}_t, \theta) p(\boldsymbol{s}_{t+1}|\boldsymbol{s}_t, \boldsymbol{a}_t) \right\}}_{\substack{\text{Markov property} \\ := p(\tau|\theta)}} \cdot \underbrace{p(\theta|D)}_{:=p_D(\theta)} \mathrm{d}\theta$$

(1)

where $\boldsymbol{s}_t$, $\boldsymbol{a}_t$, $\tau = \{(\boldsymbol{s}_t, \boldsymbol{a}_t)\}_{t=0}^{T-1}$ and $D = \{(\boldsymbol{s}_t, \boldsymbol{a}_t, \boldsymbol{s}_{t+1})\}$ are states, actions, a trajectory and observed training dataset. $\mathcal{O}_t = \{\mathcal{O}_{r,t}, \mathcal{O}_{g,t}\} \in \{0, 1\}$ represents an additional binary variable of the optimality for $(\boldsymbol{s}_t, \boldsymbol{a}_t)$ in PGM Levine (2018); Okada & Taniguchi (2020). $\mathcal{O}_{r,t} = 1$ and $\mathcal{O}_{g,t} = 1$ signify that the trajectory $\tau$ is optimized and compliant with the constraints, respectively.

In Equation 1, we can deconstruct the various components: **the probability** $p(\boldsymbol{a}_t|\boldsymbol{s}_t, \theta)$ signifies the stationary policy $\pi$ which maps one state $\boldsymbol{s}_t$ to one action $\boldsymbol{a}_t$, where $\boldsymbol{a}_t \sim p(\cdot|\boldsymbol{s}_t, \theta) = \pi(\cdot|\boldsymbol{s}_t)$ at each time step $t$; **the transition probability** $p(\boldsymbol{s}_{t+1}|\boldsymbol{s}_t, \boldsymbol{a}_t)$ represents state transitions (also known as forward-dynamics models), where $\boldsymbol{s}_{t+1} \sim p(\cdot|\boldsymbol{s}_t, \boldsymbol{a}_t)$ Chua et al. (2018) at each time step $t$; **the prior probability** $p_D(\theta)$ is derived from the posterior probability $p(\theta|D)$, where the parameter $\theta$ is inferred from the training dataset $D$; and lastly, **the optimality likelihood** $p(\mathcal{O}|\tau)$ is defined in relation to the expected reward and utility formulation of several trajectories, expressed as $\mathcal{F}_r \cdot p(\mathcal{O}_r|\tau) := \widetilde{r}(\tau)$ and $\mathcal{F}_g \cdot p(\mathcal{O}_g|\tau) := \widetilde{g}_i(\tau)$, where the operator family $\mathcal{F} = \{\mathcal{F}_r, \mathcal{F}_g\}$ and the optimality family $\mathcal{O} = \{\mathcal{O}_r, \mathcal{O}_g\}$ establish this relationship. In **Section 4.1** and **Section 5**, we offer theoretical understanding to illustrate how such specific definitions influence the RL's global convergence.

**Constrained Reinforcement Learning as Probabilistic Graphical Models.** Specifically, we consider a Constrained Markov Decision Process (CMDP) Altman (1999), a formal framework for constrained RL, which is formulated as a discounted Markov decision process with additional constrained objectives, i.e., a tuple $\langle S, A, P, R, G, \gamma \rangle$: $S$ is a finite set of states $\{\boldsymbol{s}\}$; $A$ is a finite set of actions $\{\boldsymbol{a}\}$; $P : S \times A \to S$ is a finite set of transition probabilities $\{p(\boldsymbol{s}'|\boldsymbol{s}, \boldsymbol{a})\}$; $R : S \times A \times S \to \mathbb{R}$ is a finite set of bounded immediate rewards $\{r\}$; $G : S \times A \times S \to \mathbb{R}$ comprises a finite collec-

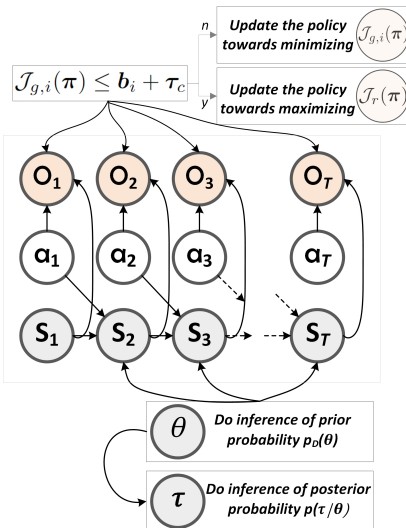

Figure 1: A new graphical model for constrained RL: refer to Algorithm 2 for a comprehensive overview of *(i) Parameter Identification*, *(ii) Policy Updating* and *(iii) Inference Execution*.

tion of unity functions $\{g\}$, where, upon satisfying the expected constraints $g_i$, the unity-optimality variable is specified as $\mathcal{O}_g = 1$; and $\gamma \in [0, 1]$ is the discount rate. A CMDP is presented as:

$$\max_{\pi} \mathcal{J}_r(\pi), \quad \text{s.t.} \quad \mathcal{J}_{g,i}(\pi) \leq \boldsymbol{b}_i + \boldsymbol{\tau}_c, \quad i = 1, ..., n \tag{2}$$

where $\mathcal{J}_r(\pi) := \mathbb{E}[\sum_{t=0}^{\infty} \gamma^t r(\boldsymbol{s}_t, \boldsymbol{a}_t)|\pi, \boldsymbol{s}_0 = s]$ and $\mathcal{J}_{g,i}(\pi) := \mathbb{E}[\sum_{t=0}^{\infty} \gamma^t g_i(\boldsymbol{s}_t, \boldsymbol{a}_t)|\pi, \boldsymbol{s}_0 = s]$ are the value function associated with the immediate reward $r$ and the utility $g$, respectively; $b_i$ is a fixed limit for the $i$-th constraint; and $\boldsymbol{\tau}_c$ is the tolerance. Figure 1 shows how constrained RL can be viewed as a novel variation of PGMs.

## 4 ADAPTIVE SLICED WASSERSTEIN VARIATIONAL OPTIMIZATION

In this section, we present AWaVO's two primary submodules: WVI and PPO-DR. The detailed algorithm is outlined in Algorithm 2, and the overarching algorithmic structure is depicted in Figure 2.

### 4.1 WVI: WASSERSTEIN VARIATIONAL INFERENCE

**Variational Inference for Dynamic Uncertainties.** Given uncertainties in a dynamics model, it is reasonable to assume that the optimal trajectories $\{\tau\}$ are uncertain. To infer optimal policies under uncertainties, let us consider a variational inference: $D(q_\theta(\tau)||p(\tau|\mathcal{O}))$, where, for simplicity, we use $p(\tau|\mathcal{O})$ to represent $p(\tau|\mathcal{O}_{t:T} = 1)$ ; and $D(\cdot||\cdot)$ represents a distance metric between two probabilities. Building upon Equation 1, the variational distribution $q_\theta(\tau)$ is constructed as $q_\theta(\tau) = q(\boldsymbol{a})p(\tau|\theta)p_D(\theta)$. The construction suggests an assumption that the state transitions are controlled by $p(\boldsymbol{s}_{t+1}|\boldsymbol{s}_t, \boldsymbol{a}_t)$. According to Equation 1, we formulate the posterior as $p(\tau|\mathcal{O}) \propto p(\mathcal{O}|\tau)p(\tau|\theta)p_D(\theta)$. Note that our implementation (**Section 6**) uses $p(\mathcal{O}|\tau)p(\tau|\theta)p_D(\theta)$.

While Kullback-Leibler (KL) divergence is widely used in conventional variational inference, its application in certain practical implementations can be risky due to its limitations, including asymmetry and infinity, arising when there are unequal supports. In this section, we extend the Wasserstein distance into the variational inference, and present the derivation of how we transform the GSWD between the two posteriors to the optimality likelihood $p(\mathcal{O}|\tau)$ and its approximation $q(\boldsymbol{a})$.

**Adaptive Generalized Sliced Wasserstein Distance.** GSWD has exhibited high projection efficiency in previous studies Kolouri et al. (2019); Chen et al. (2020) (please refer to **Appendix A** for a comprehensive background and definition of Wasserstein distance). However, the identification of the hypersurface hyperparameters, such as $l$ and $\widetilde{\theta}$, remains to be a challenge. The selection of these parameters, specifying the hypersurface along with its slicing direction, is generally a task-specific problem and requires prior knowledge or domain expertise. We now present a novel adaptive sliced Wasserstein distance, called A-GSWD, that integrates GSWD with PPO-DR, an adaptive process for determining the parameters of a hypersurface. Following GSWD's definition (Equation 6 in **Appendix A.2**), we introduce the definition of A-GSWD by utilizing PPO-DR for the adaptive slicing:

$$\mathbf{A} - \mathbf{GSWD}_k(\mu, \nu) = \left( \int_{\mathcal{R}_{\widetilde{\theta}}} W_k^k \left( \mathcal{G}_\mu \left( \cdot, \widetilde{\theta}; g_{rl} \right), \mathcal{G}_\nu \left( \cdot, \widetilde{\theta}; g_{rl} \right) \right) \mathrm{d}\widetilde{\theta} \right)^{\frac{1}{k}} \tag{3}$$

where the push-forward operator $\mathcal{G}_\mu(l, \widetilde{\theta}) = \int_{\mathbb{G}^d} \delta(l - \langle x, \widetilde{\theta} \rangle) \mathrm{d}\mu$. $l \in \mathbb{R}$ and $\widetilde{\theta} \in \mathcal{R}_{\widetilde{\theta}}$ represent the parameters of hypersurfaces, both of which are the outputs from actor networks in PPO-DR. $\mathcal{R}_{\widetilde{\theta}} \subset \mathbb{R}^d$ is a compact set of all feasible parameters $\widetilde{\theta}$, where $\mathcal{R}_{\widetilde{\theta}} = \mathbb{S}^{d-1}$ for $g_{rl}(\cdot, \widetilde{\theta}) = \langle \cdot, \widetilde{\theta} \rangle$. Although the proposed adaptive slicing method, i.e., A-GSWD, improves the efficiency and accuracy of the Wasserstein distance computation, its demonstration on a valid metric guarantee remains a problem Kolouri et al. (2019). In **Section 5**, we prove that the proposed A-GSWD is a true metric that satisfies non-negativity, symmetry, the triangle inequality and $\mathbf{A} - \mathbf{GSWD}_k(\mu, \mu) = 0$, respectively.

We then employ A-GSWD to address the variational inference, i.e., minimizing the distance $D(q_\theta(\tau)||p(\tau|\mathcal{O})) = \mathbf{A} - \mathbf{GSWD}_k(q_\theta(\tau), p(\tau|\mathcal{O}))$ between the variational distribution $q_\theta(\tau)$ and the posterior distribution $p(\tau|\mathcal{O})$. Subsequently, the variational inference can be reformulated to the minimization problem, as shown in WVI of Figure 2: $\arg\min_{q_\theta(\tau)} \mathbf{A} - \mathbf{GSWD}_k(q(\boldsymbol{a}), p(\mathcal{O}|\tau))$, where $p(\mathcal{O}|\tau)$ represents the optimality likelihood, and the detailed derivation is in **Appendix C.1**.

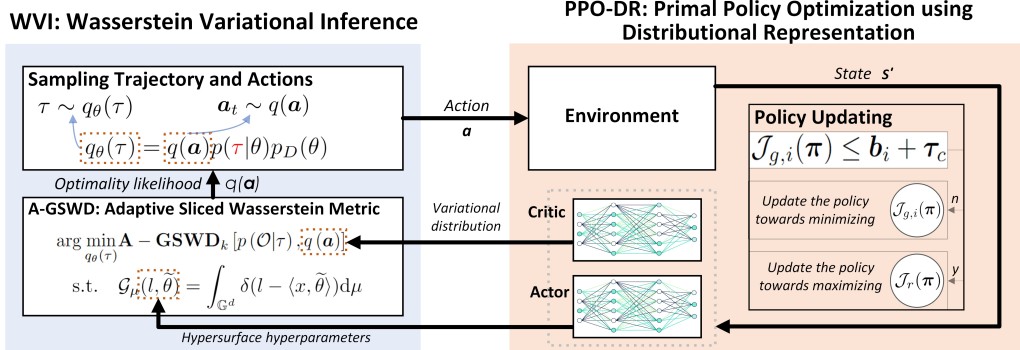

Figure 2: The algorithmic framework of AWaVO. We reform constrained RL as a Wasserstein variational optimization setup, consisting of two primary submodules: PPO-DR and WVI (**Section 4**).

## 4.2 PPO-DR: Primal Policy Optimization using Distributional Representation

The current policy optimization for constrained RL can be classified into two categories: primal-dual and primal approaches Xu et al. (2021). The former, transforming the constrained problem into an unconstrained one, are most commonly used although sensitive to Lagrange multipliers and other hyperparameters, such as the learning rate. On the other hand, the latter (i.e., primal approaches) require less hyperparameter tuning but have received less attention in terms of convergence demonstration compared to the primal-dual approaches.

**Policy Optimization combining Optimality Likelihood**  Based on the definition in **Section 4.1**, a constrained RL problem, as outlined in Equation 2, can be iteratively substituted and resolved as:

$$
\begin{cases}
\arg\max_{q(\boldsymbol{a})} \mathbb{E}[\mathcal{F}_r \cdot p\left(\mathcal{O}_r|\tau\right)], & \mathcal{J}_{g,i}(\pi) \leq \boldsymbol{b}_i + \boldsymbol{\tau}_c \\
\arg\min_{q(\boldsymbol{a})} \mathbb{E}[\mathcal{F}_g \cdot p\left(\mathcal{O}_g|\tau\right)], & otherwise
\end{cases}
\tag{4}
$$

where we recall that $\{\mathcal{F}_r, \mathcal{F}_g\}$ are two operators defined as $\mathcal{F}_r \cdot p\left(\mathcal{O}_r|\tau\right) := \widetilde{r}(\tau)$ and $\mathcal{F}_g \cdot p\left(\mathcal{O}_g|\tau\right) := \widetilde{g}_i(\tau)$, respectively. Furthermore, we can calculate the accumulated reward and utility function as $\widetilde{r}(\tau) = \mathbb{E}\big[\sum_{t=0}^{T-1} \gamma^t r(\boldsymbol{s}_t, \boldsymbol{a}_t)\big]$ and $\widetilde{g}_i(\tau) = \mathbb{E}\big[\sum_{t=0}^{T-1} \gamma^t g_i(\boldsymbol{s}_t, \boldsymbol{a}_t)\big]$, respectively. Consequently, we obtain $\mathcal{J}_r(\pi) = \mathbb{E}[\widetilde{r}(\tau)]$ and $\mathcal{J}_{g,i}(\pi) = \mathbb{E}[\widetilde{g}_i(\tau)]$ if $T = \infty$.

If we only define $\mathcal{F}_r \propto \log[\cdot]$, it becomes equivalent to the formulation used in Levine (2018); Okada & Taniguchi (2020; 2018). In this case, we can retrieve an optimization process that resembles Model Predictive Path Integral (MPPI) Okada & Taniguchi (2018). The design of reward functions in the traditional RL is typically based on task-specific heuristics, and is often considered as much an art as science. We will present such interpretation in **Section 5** to show how the reward operator family $\mathcal{F}$ acts on convergence, as well as a more rigorous approach to ensure guaranteed global convergence rate during the training process. Additionally, in **Section 6**, we empirically verify these theoretical guarantees.

**Policy Updating.**  As shown in Algorithm 1, we first update the policy towards either maximizing $\mathcal{J}_r(\pi)$ or minimizing $\mathcal{J}_{g,i}(\pi)$ by using the distributional representation (introduced in **Appendix B**), where the gradient of actor and critic network, denoted as $\delta_{\theta^\mu}$ and $\delta_{\theta^Q}$, are defined in Equation 10 in **Appendix B**. Then, as shown in PPO-DR of Figure 2, the actor network generates the parameters of hypersurfaces for adaptively selecting the slicing directions and hypersurfaces to improve the accuracy of the Wasserstein distance computation; and the critic network provides an entire state-action distribution, which is directly utilized as the variational distribution of the optimality likelihood $q(\boldsymbol{a})$ in A-GSWD (i.e., Equation 3), as shown in Figure 2.

## 5 Formal methods for learning interpretability

**Proposition 1.** *(Metric)*: Given two probability measures $\mu, \nu \in P_k(\mathcal{X})$, and a mapping $g_{rl}: \mathcal{X} \to \mathcal{R}_{\widetilde{\theta}}$, A-GSWD defined in Equation 3 of order $k \in [1, \infty)$ is a true metric that satisfies non-negativity,

---

**Algorithm 1** PPO-DR: Primal Policy Optimization using Distributional Representation

---

**Input:** $s_k, s_{k+1}, \tau_c, \theta^\mu, \theta^Q$
**Output:** updated $\theta^\mu, \theta^Q$
1: **Constraint Estimation**: estimate the constraints: $\mathcal{J}_{g,i}(\pi_\theta(s, a)) = \mathbb{E}[\widetilde{g}_i(\tau)], \; \forall i \in [1, p]$
2: **Policy Improvement**:
3: **if** $\mathcal{J}_{g,i}(\pi) \leq b_i + \tau_c, \forall i \in [1, p]$ **then**
4:     Update the policy towards maximizing $\mathcal{J}_r(\pi)$: $\theta^\mu \leftarrow \theta^\mu + l_\mu \delta_{\theta^\mu}$, and $\theta^Q \leftarrow \theta^Q + l_\theta \delta_{\theta^Q}$
5: **else**
6:     Update the policy towards minimizing $\mathcal{J}_{g,i}(\pi)$: $\theta^\mu \leftarrow \theta^\mu - l_\mu \nabla_{\theta^\mu} \widetilde{g}_i(\tau)$, and $\theta^Q \leftarrow \theta^Q - l_Q \nabla_{\theta^Q} \widetilde{g}_i(\tau)$
7: **end if**

---

**Algorithm 2** AWaVO: Adaptive Sliced Wasserstein Variational Optimization

---

**Input:** $s_k, s_{k+1}, \theta^\mu, \theta^Q$
**Output:** $a_k$
1: **Initialize**:
    $\theta = [\theta^\mu, \theta^Q]$: the parameters of actor and critic network
2: **Repeat**
3: **for** $t = 0, 1, 2, ..., T - 1$ **do**
4:     **Parameter Identification:** achieve $p_D(\theta)$ by doing inference of the posteriors $p(\theta|D)$ (Section 3)
5:     **Policy Updating:** $\{\theta^\mu, \theta^Q\} \leftarrow$ Exec. Algorithm 1 $(s_k, s_{k+1}, \tau_c, \theta^\mu, \theta^Q)$
6:     **Inference Execution:** do inference of the posterior probability, as described in Section 4.1
        $p(\tau|\mathcal{O}_{t:T}) \leftarrow \arg \min_{q_\theta(\tau)} \mathbf{A} - \mathbf{GSWD}_k \left( q(a), p(\mathcal{O}|\tau) \right)$
7:     sample actions $a_k \leftarrow p(\tau|\mathcal{O}_{t:T})$, execute $a_k$, and observe $s_{k+1}$
8: **end for**

---

symmetry, triangle inequality and $\mathbf{A} - \mathbf{GSWD}_k(\mu, \mu) = 0$, *if and only if* $g_{rl}$ in Equation 3 is an injective mapping. See **Appendix C.3** for Proof.

**Remark 1.** If the mapping $g_{rl}$ lacks injectivity, A-GSWD is still considered as a pseudo-metric, which maintains significant properties including non-negativity, symmetry and triangle inequality.

To establish a link between the reward operator family $\mathcal{F}$ and the global convergence of PPO-DR, here, we first introduce the **Conditions** and then present **Theorem 1** *(Global Convergence)*.

**Conditions.**     Regarding $\mathcal{F}$, the reward operator family: *(i)* $\mathcal{F}_r$ is monotonically increasing and continuously defined on $(0, 1]$, and the range covers $[r_{\min}, r_{\max}]$; and *(ii)* $\mathcal{F}_g$ is monotonically decreasing and continuously defined on $(0, 1]$, and the range covers $[r_{\min}, r_{\max}]$.

**Theorem 1.** *(Global Convergence)*: Given the policy in the $i$-th policy improvement $\pi^i$, $\pi^i \to \pi^*$ and $i \to \infty$, there exists $Q^{\pi^*}(s, a) \geq Q^{\pi^i}(s, a)$ *if and only if* the reward operator family $\mathcal{F}$ satisfies the both **Conditions**. See **Appendix C.3** for Proof.

Next we demonstrate a more rigorous understanding of how the precise definition of $\mathcal{F}$ impacts the convergence rate. As far as we know, this is the first attempt to gain an inherent understanding of how the reward design affects convergence in RL.

**Theorem 2.** *(Global Convergence Rate)*: Let $m$ and $H$ be the width and layers of a neural network, $K_{td} = (1 - \gamma)^{-\frac{3}{2}} m^{\frac{H}{2}}$ be the iterations required for convergence of the distributional Temporal Difference (TD) learning (defined in Equation 12), $l_Q = \frac{1}{\sqrt{T}}$ be the policy update (in *Line 4* of Algorithm 1) and $\tau_c = \Theta(\frac{1}{(1-\gamma)\sqrt{T}}) + \Theta(\frac{1}{(1-\gamma)Tm^{\frac{H}{4}}})$ be the tolerance (in *Line 3* of Algorithm 1).

There exists a global convergence rate of $\Theta(1/\sqrt{T})$, and a sublinear rate of $\Theta(1/\sqrt{T})$ if the constraints are violated with an error of $\Theta(1/m^{\frac{H}{4}})$, with probability of at least $1 - \delta$. This holds *if and only if* the reward operator family $\mathcal{F}$ satisfies both **Conditions**. See **Appendix C.3** for Proof.

**Probabilistic interpretation on sequential decisions.**     We now quantitatively establish the relationships between latent factors, such as disturbances, that possibly influence decision-making and the sequential decisions, namely trajectories, by providing a probabilistic interpretation. Referring to the abbreviation presented in Equation 1, we reform it as: $p(\tau|D) = p(\mathcal{O}|\tau) \cdot p(s, a|\theta) \cdot p_D(\theta)$. Then, the latent factors are denoted by $L = \{L_i\}_{i=0}^{M-1}$, where $M$ represents the total num-

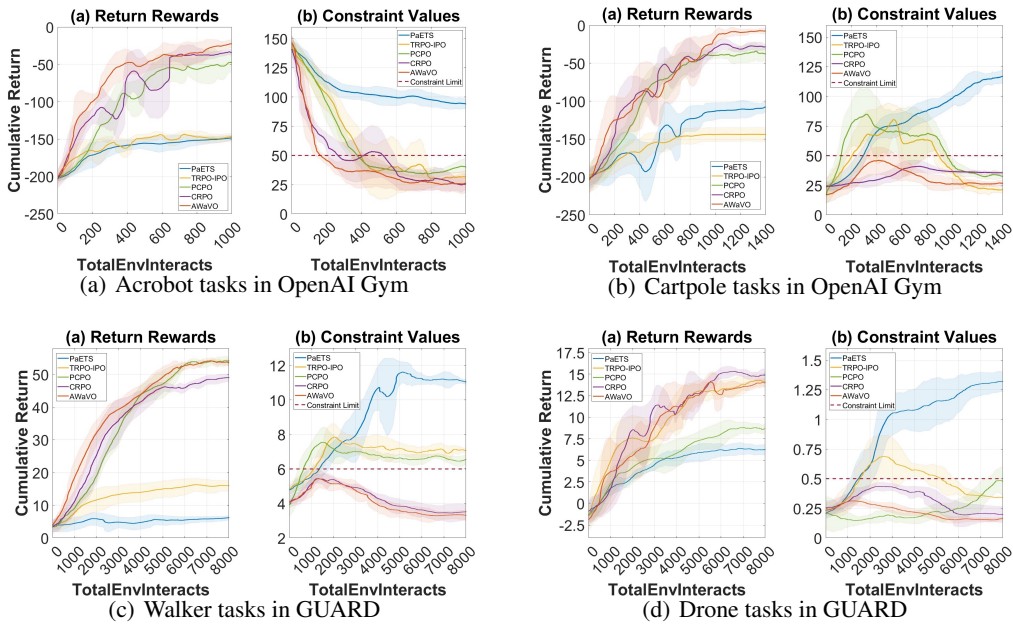

Figure 3: Performance comparison over 10 seeds. CRPO and AWaVO outperform PaETS, with a trade-off highlighted: although PaETS offers probabilistic interpretation with Bayesian networks, its convergence is generally unstable. Our proposed AWaVO achieves a better balance in high performance and interpretability. In contrast to two other constrained RL algorithms, i.e., TRPO-IPO and PCPO, we observe an interesting result: PCPO performs better in tasks like Acrobot, Cartpole, and Walker, while TRPO-IPO outperforms PCPO in the more complex drone tasks (Figure 3(d)). Further, in Figure 6, we will explore more complex real-world tasks using an aerial robot.

ber of defined factors. By applying the chain rule to the posterior probability $p(\tau|D)$, we have $\{p(\tau|L_i)\}_{i=0}^M = \left\{ \frac{p(\tau|D)}{p(L_i|D)} \right\}_{i=0}^M$, where the equation provides a decomposition of the joint posterior probability $p(\tau|D)$ into conditional probabilities that involve individual factors $L_i$. Practically, this decomposition enables a probabilistic interpretation of each factor's impact on the policy. While the theoretical simplicity of this decomposition is noteworthy, its practical significance is particularly evident in real-world safety-critical applications, such as robot autonomy. In **Section 6**, we showcase numerical examples to illustrate such probabilistic interpretation.

## 6 EXPERIMENTS

In this section, we conduct empirical assessments of AWaVO's performance in both simulated platforms and real-world robot tasks. Initially, we perform tasks with multiple constraints in OpenAI Gym framework Brockman et al. (2016). Then we showcase AWaVO's practicality through real quadrotor Flight Tasks (FTs), which provides a more comprehensive assessment of its performance. These evaluations serve a dual purpose: to validate AWaVO's performance; and, critically, to empirically demonstrate its quantitative interpretability. This interpretability includes confirming properties such as the guaranteed convergence rate as demonstrated in **Theorem 2** and the probabilistic decision interpretation discussed in **Section 5** within the context of sequential decision-making tasks.

**Comparative Performance in Simulated Tasks.** Here we conduct tasks with multiple constraints in OpenAI Gym framework Brockman et al. (2016) and GUARD Zhao et al. (2023), a safe RL benchmark: Acrobot, Cartpole, Walker and Drone. We use four constrained RL as our benchmark approaches: PaETS Okada & Taniguchi (2020), i.e., a Bayesian RL combining with variational inference, TRPO-IPO Liu et al. (2020), i.e., an enhanced variant of TRPO-Lagrangian Bohez et al. (2019), PCPO Yang et al. (2020), i.e., an advanced variant of CPO Achiam et al. (2017) and CRPO Xu et al. (2021), i.e., a primal constrained RL approach.

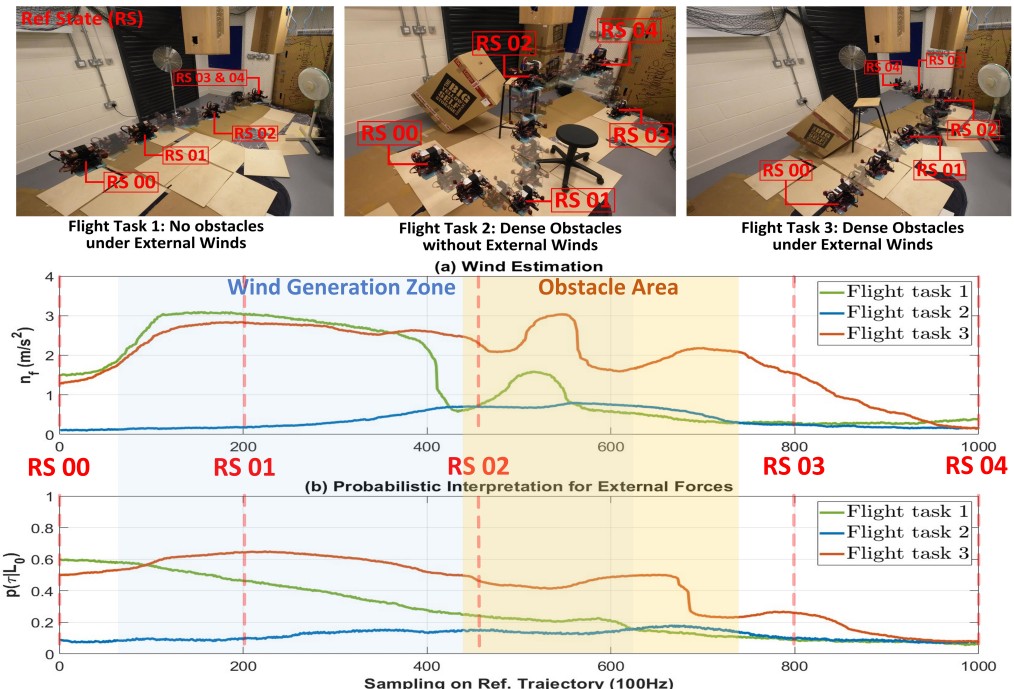

Figure 4: Probabilistic interpretation of sequential decisions. We empirically demonstrate this interpretation through the execution of three real-world FTs. Here, **the probability** $p(\tau|L_0)$ reveals the degree to which the measurement of external forces Ding et al. (2021), denoted as $n_f$, influences the quadrotor's sequential decisions. For instance, in the case of 'RS 02', situated in an area with a mix of wind and obstacles, both aerodynamic effects (i.e., external forces) from winds and obstacles act on the quadrotor concurrently. Quantitatively, the red $p(\tau|L_0)_{FT3}$ is approximately equal to the sum of $p(\tau|L_0)_{FT1}$ (only wind) and $p(\tau|L_0)_{FT2}$ (only obstacles).

The parameter setting of AWaVO is shown in Table 1 of **Appendix D.1**, which is based on our benchmarks, i.e., CRPO Xu et al. (2021) and GUARD Zhao et al. (2023). According to our proposed **Proposition 1** and the **Proposition 1** presented in Kolouri et al. (2019), the defining function $g_{rl}(\cdot, \widetilde{\theta})$ can be defined as homogeneous polynomials, i.e., $g_{rl}(\cdot, \widetilde{\theta}) = \sum_{|\alpha|=m} \widetilde{\theta}_{\alpha} x^{\alpha}$, where the defining function $g_{rl}$ is injective if the degree of the polynomial $m$ is odd. Thus we set $m = 3$ based on Kolouri et al. (2019). The comprehensive task descriptions are available in **Appendix D.2**.

As per the benchmarks provided by CRPO Xu et al. (2021) and GUARD Zhao et al. (2023), the training process comprises 1000 iterations for the Acrobot and Cartpole tasks and 6000 iterations for the drone and walker tasks, respectively. We establish the constraint limit to facilitate a straightforward comparison of constraint convergence; see **Appendix D.2** for additional details. The tolerance is set as $\tau_c = 0.5$, following CRPO Xu et al. (2021). By analyzing the comparative training performances in Figure 3, we observe that the superiority of CRPO and AWaVO stems from their primal safe RL nature, which involves training under constraints and ensuring global convergence rate. Although CRPO exhibits comparable or slightly better convergence performance than AWaVO, as evident in Figure 3(d), we place greater emphasis on two other aspects: training convergence under uncertainties and decision-making interpretation. In Figure 6 below, we provide comparative demonstrations in real robot tasks to showcase how AWaVO effectively balances a trade-off between performance and interpretability in a more complex sequential decision-making scenario.

Furthermore, we empirically verify the formal method **Theorem 2** *(Global Convergence Rate)* on the convergence rate, and conclude that, based on the average performance, the convergence rate of AWaVO is in the range of $\Theta(1/\sqrt{T}) < C_{rate} \leq \Theta(1/T^{1.2})$. According to the results shown in Figure 3(d), CRPO performs better than our AWaVO in the simulated drone task, with the absence of disturbances. Subsequently, in Figure 6 below, we further evaluate these approaches in a real-world physical environment characterized by varying uncertainties, leading to different outcomes. It is

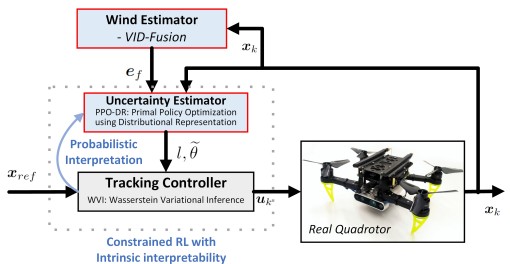
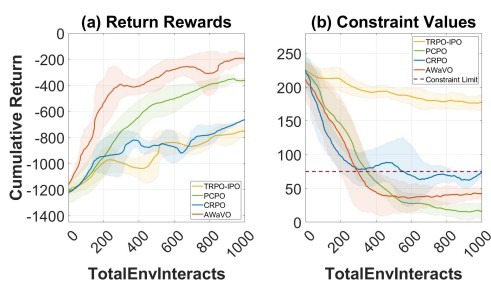

Figure 5: We use our AWaVO as the tracking controller for a quadrotor, where PPO-DR is employed as the uncertainty estimator, and WVI using A-GSWD is leveraged as the controller.

Figure 6: Performance comparison in a real quadrotor: our AWaVO slightly outperforms the constrained RL approach, i.e., PCPO, whilst achieving interpretability in Figure 4.

worth noting that our approach incorporates two optimizations for handling uncertainties: variational inference and policy updating. This combination reduces the frequency of policy updates whilst enhancing our ability to handle uncertainties. In the upcoming real robot task, we will introduce variable disturbances to demonstrate our capability to optimize policies under uncertain conditions.

**Comparative Performance in Real-world Tasks.** We demonstrate the effectiveness of our proposed AWaVO by practically implementing it in real-world decision-making problems. The tracking control framework, shown in Figure 5, is an end-to-end learning-based framework. The tasks' aim is to track the reference effectively and accurately, where VID-Fusion Ding et al. (2021) is used to measure external forces such as aerodynamic effects. The specific FTs are shown in Figure 4, where FT 1 - tracking reference trajectories under external forces without obstacles; FT 2 - tracking trajectories around the static (no external forces) but dense obstacles; and FT 3 - tracking trajectories under external forces around dense obstacles. The technical specification of the quadrotor is shown in Table 2 of **Appendix D.1**. The training convergence is demonstrated in Figure 6. To view the hardware experiments in action, please refer to the accompanying video demonstration [1].

Next, we illustrate the interpretation of sequential decisions, i.e., the actual control commands fed into the four motors. Leveraging the Intel RealSense D435i depth camera onboard, we can detect obstacles and estimate external forces. These latent factors, denoted as $L = L_0, L_1$, represent external forces and obstacles, respectively. The probability $p(\tau|L)$ reveals why the quadrotor makes these decisions and quantifies the extent to which factor $L$ contributes to the sequential decisions, i.e., $\tau$. Figure 4 presents a quantitative interpretation, i.e., $p(\tau|L_0)$, indicating the magnitude and evolution that the external force $n_f$ impacts on the current control decisions.

Practically, this probabilistic interpretation represents significant progress in addressing a longstanding and challenging question: why do the machine systems powered by Artificial Intelligence (AI) technologies make certain decisions, and what are the exact latent factors influencing those decisions? Such progress holds particular value for safety-critical industries like self-driving vehicles, aerospace engineering and high-frequency trading in financial services, particularly in cases where AI-based approaches exhibit erratic performance and thorough analysis is necessary.

## 7 CONCLUSION & LIMITATION

Enthusiasm towards the possible applications of RL is growing worldwide. Lacking sufficient ability to interpret an agent's actions and its policy optimizations, however, makes it infeasible to deploy RL in safety-critical domains like advanced manufacturing, autonomous systems and financial trading. Our primary motivation in introducing AWaVO, an intrinsically interpretable RL framework, is to tackle key interpretability challenges concerning convergence guarantees, optimization transparency, and sequential-decision interpretation. Empirical results demonstrate that the proposed AWaVO balances a reasonable trade-off between high performance and quantitative interpretability in both simulation and real-world robotic tasks. The primary limitation we encounter is ensuring the trustworthiness of the posterior probability generated by the critic network, which operates as a Bayesian network. Our ongoing efforts involve applying statistical methods to establish a specific confidence interval for the Bayesian network's outcomes, and implementing AWaVO in several safety-critical applications to demonstrate its effectiveness in additional real-world scenarios.

## 8 REPRODUCIBILITY STATEMENT

Enhancing interpretability in reinforcement learning aligns with the broader goal of improving reproducibility within the RL community. Therefore, from this standpoint, the accomplishments outlined in this paper, especially in terms of training transparency associated with the reward function design, significantly contribute to enhancing the reproducibility of RL. Specially, to help readers reproduce our experiments, we have detailed our architectural designs in Section 4, and implementation details in Table 1 and Table 2 of Appendix D.1. Our code will be made openly available upon acceptance of our manuscript for publication. Additionally, we provide the video link to demonstrate the real-world quadrotors experiments.

## 9 ETHICS STATEMENT

The authors anticipate no adverse ethical repercussions from their work and declare no conflicts of interest.

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

# Appendix

## A  BACKGROUND ON WASSERSTEIN DISTANCE

### A.1  SLICED WASSERSTEIN DISTANCE

A fundamental challenge in both machine learning and statistics communities is to form effective metrics between pairs of probability distributions. Weaker notions, such as divergence measures, including KL divergence Kullback & Leibler (1951), have been proposed and widely used. However, such measures do not satisfy the two basic properties of a metric, namely symmetry and triangle inequality. To address this issue, interest has rapidly increased in optimal transport in recent years. In this subsection, we introduce the Wasserstein distance and its variants, including SWD Rabin et al. (2012); Nietert et al. (2022) and GSWD Kolouri et al. (2019), as metrics that conditionally satisfy the properties.

Let $\Gamma(\mu, \nu)$ be a set of all transportation plans $\gamma \in \Gamma(\mu, \nu)$, where $\gamma$ is a joint distribution over the space $\mathcal{X} \times \mathcal{X}$, and $\mu, \nu \in P_k(\mathcal{X})$ are two measures of probability distributions over $\mathcal{X}$. $P_k(\mathcal{X})$ represents a set of Borel probability measures with finite k-th moment on a Polish metric space Villani et al. (2009). $d(x, y)$ represents a distance function over $\mathcal{X}$. The Wasserstein distance of order $k \in [1, \infty)$ between two measures $\mu, \nu$ is defined as Villani et al. (2009): $W_k(\mu, \nu) = \left( \inf_{\gamma \in \Gamma(P,Q)} \int_{\mathcal{X} \times \mathcal{X}} d(x, y)^k \mathrm{d}\gamma(x, y) \right)^{1/k}$. This definition, however, involves solving an optimization problem that is computationally expensive in practical implementation, particularly for high-dimensional distributions. Thus sliced $k$-Wasserstein distance Rabin et al. (2012); Nietert et al. (2022), defined over spaces of hyperplanes in $\mathbb{R}^d$, is proposed as a computationally efficient approximation:

$$\mathbf{SWD}_k(\mu, \nu) = \left( \int_{\mathbb{S}^{d-1}} W_k^k \left( \mathcal{R}_\mu \left( \cdot, \widetilde{\theta} \right), \mathcal{R}_\nu \left( \cdot, \widetilde{\theta} \right) \right) \mathrm{d}\widetilde{\theta} \right)^{\frac{1}{k}} \tag{5}$$

where Radon transform $\mathcal{R}$ Radon (2005) is introduced in SWD to map a function $f(\cdot)$ to the hyperplanes $\left\{ x \in \mathbb{R}^d | \langle x, \widetilde{\theta} \rangle = l \right\}$, i.e., $\mathcal{R}f(l, \widetilde{\theta}) = \int_{\mathbb{R}^d} f(x)\delta(l - \langle x, \widetilde{\theta} \rangle)\mathrm{d}x$: $l \in \mathbb{R}$ and $\widetilde{\theta} \in \mathbb{S}^{d-1} \subset \mathbb{R}^d$ represent the parameters of these hyperplanes. In the definition of SWD, the Radon transform $\mathcal{R}_\mu$ is employed as the push-forward operators, defined by $\mathcal{R}_\mu(l, \widetilde{\theta}) = \int_{\mathbb{R}^d} \delta(l - \langle x, \widetilde{\theta} \rangle)\mathrm{d}\mu$ Kolouri et al. (2019).

### A.2  GENERALIZED SLICED WASSERSTEIN DISTANCE

While SWD offers a computationally efficient way to approximate the Wasserstein distance, the projections are limited to linear subspaces, such as hyperplanes $\{x\}$. Due to the nature of these linear projections, the resulting metrics typically have low projection efficiency in high-dimensional spaces Kolouri et al. (2019); Deshpande et al. (2019). Thus various variants of SWD are proposed to enhance its projection effectiveness. Specifically, the GSWD Kolouri et al. (2019), defined in Equation 6, is proposed by incorporating nonlinear projections. Its main novelty is that Generalized Radon Transforms (GRTs) $\mathcal{G}$ Beylkin (1984); Ehrenpreis (2003); Homan & Zhou (2017), i.e., $\mathcal{G}f(l, \widetilde{\theta}) = \int_{\mathbb{R}^d} f(x)\delta(l - \beta(x, \widetilde{\theta}))\mathrm{d}x$, are used to define the nonlinear projections towards hypersurfaces rather than linear projections to the hyperplanes in SWD. Let $\beta(x, \widetilde{\theta})$ be *a defining function* when satisfying the conditions **H.1-H.4** in Kolouri et al. (2019).

$$\mathbf{GSWD}_k(\mu, \nu) = \left( \int_{\mathcal{X}_{\widetilde{\theta}}} W_k^k \left( \mathcal{G}_\mu \left( \cdot, \widetilde{\theta} \right), \mathcal{G}_\nu \left( \cdot, \widetilde{\theta} \right) \right) \mathrm{d}\widetilde{\theta} \right)^{\frac{1}{k}} \tag{6}$$

where $\widetilde{\theta} \in \mathcal{X}_{\widetilde{\theta}}$ and $\mathcal{X}_{\widetilde{\theta}}$ is a compact set of all feasible parameters $\widetilde{\theta}$ for $\beta(\cdot, \widetilde{\theta})$, e.g., $\mathcal{X}_{\widetilde{\theta}} = \mathbb{S}^{d-1}$ for $\beta(\cdot, \widetilde{\theta}) = \langle \cdot, \widetilde{\theta} \rangle$. The GRT operator $\mathcal{G}_\mu$ is utilized as the push-forward operator, i.e., $\mathcal{G}_\mu(l, \widetilde{\theta}) = \int_{\mathbb{G}^d} \delta(l - \langle x, \widetilde{\theta} \rangle)\mathrm{d}\mu$. For the theoretical properties of a metric, SWD is a true metric that satisfies both symmetry and triangle inequality Bonnotte (2013), where the approximation error is obtained and analyzed in Nadjahi et al. (2020). The GSWD defined by Equation 6 is a true metric if and only if $\beta(\cdot)$ in $\mathcal{G}$ is a injective mapping Chen et al. (2020).

## B  BACKGROUND ON DISTRIBUTIONAL REPRESENTATION IN BELLMAN EQUATION AND TEMPORAL DIFFERENCE LEARNING

### B.1  REASONING BEHIND DISTRIBUTIONAL REPRESENTATION

The motivation for employing a distributional representation is twofold. Firstly, it provides a more comprehensive and richer value-distribution information, thereby enhancing the stability of the learning process. This stability is particularly important for Bayesian learning processes, which often encounter challenges in achieving stable convergence. Secondly, the distributional representation contributes significantly to interpretability. As illustrated in Equation 18 of the proof, it uses quantiles derived from the distributional representation to formally establish the transparency of convergence process outlined in **Theorem 2**.

### B.2  DISTRIBUTIONAL REPRESENTATION IN BELLMAN EQUATION

Unlike traditional RL, where the primary objective is to maximize the expected value-action function $Q$, the distributional Bellman equation Bellemare et al. (2017) was proposed to approximate and parameterize the entire distribution of future rewards. In the setting of policy evaluation, given a deterministic policy $\pi$, the *Bellman operator* $\mathcal{T}^\pi$ is defined as Bellemare et al. (2017); Dabney et al. (2018):

$$\mathcal{T}^\pi Z(\boldsymbol{s}, \boldsymbol{a}) \overset{D}{:=} R(\boldsymbol{s}, \boldsymbol{a}) + \gamma Z(S', A') \tag{7}$$

where $Z^\pi$ denotes the state-action distribution, and $R(\boldsymbol{s}, \boldsymbol{a})$ denotes the reward distribution. In control setting, a distributional *Bellman optimality operator* $\mathcal{T}$ with quantile approximation is proposed in Dabney et al. (2018):

$$\mathcal{T} Z(\boldsymbol{s}, \boldsymbol{a}) \overset{D}{:=} R(\boldsymbol{s}, \boldsymbol{a}) + \gamma Z(\boldsymbol{s'}, \arg\max_{a'} \mathbb{E}_{\boldsymbol{p}, R}[Z(\boldsymbol{s'}, \boldsymbol{a'})]) \tag{8}$$

where we let $Z_\theta := \frac{1}{N} \sum_{i=1}^N \delta_{q_i(\boldsymbol{s}, \boldsymbol{a})}$ be a quantile distribution mapping one state-action pair $(\boldsymbol{s}, \boldsymbol{a})$ to a uniform probability distribution supported on $q_i$. Based on Equation 7, a contraction is demonstrated Dabney et al. (2018) over the Wasserstein metric: $\bar{d}_\infty(\Pi_{W_1}\mathcal{T}^\pi Z_1, \Pi_{W_1}\mathcal{T}^\pi Z_2) \leq \bar{d}_\infty(Z_1, Z_2)$, where $\bar{d}_k := \sup W_k(Z_1, Z_2)$ denotes the maximal form of the $k$-Wasserstein metrics. $W_k$, $k \in [1, \infty]$ denotes the $k$-Wasserstein distance. $\Pi_{W_1}$ is a quantile approximation under the minimal 1-Wasserstein distance $W_1$.

### B.3  DISTRIBUTIONAL REPRESENTATION IN TEMPORAL DIFFERENCE LEARNING

Building upon the aforementioned contraction guarantees, we utilize distributional TD learning to estimate the distribution of state-action value, denoted as $Z$. In each iteration, we have the following:

$$\begin{aligned}\zeta_{k+1}^i(\boldsymbol{s}, \boldsymbol{a}) &= \zeta_k^i(\boldsymbol{s}, \boldsymbol{a}) + l_{td}\boldsymbol{\Delta}_k^i \\ &= \zeta_k^i(\boldsymbol{s}, \boldsymbol{a}) + l_{td} \times \bar{d}_\infty(\Pi_{W_1}\mathcal{T}^\pi(h_i(\boldsymbol{s}, \boldsymbol{a}, \boldsymbol{s'}) + \gamma\zeta_k^i(\boldsymbol{s'})), \Pi_{W_1}\mathcal{T}^\pi\zeta_k^i(\boldsymbol{s}, \boldsymbol{a}))\end{aligned} \tag{9}$$

where $\zeta_k^i \in S \times A$ represents the estimated distribution of the state-action distribution $Z$ in the $k$-th TD-learning-iteration for all $i = 0, ..., p$. The TD learning rate is denoted as $l_{td}$. The function $h_i : S \times A \times S \to \mathbb{R}$ maps the triple $(\boldsymbol{s}, \boldsymbol{a}, \boldsymbol{s'})$ to a *real number*. Specifically, $h_i$ is defined as $h_i = r$ when $i = 0$; and $h_i = g_i$ when $i \in [1, n]$. The distributional TD error $\boldsymbol{\Delta}_k^i$ in Equation 12 is calculated by $\bar{d}_\infty(\Pi_{W_1}\mathcal{T}^\pi(h_i(\boldsymbol{s}, \boldsymbol{a}, \boldsymbol{s'}) + \gamma\zeta_k^i(\boldsymbol{s'})), \Pi_{W_1}\mathcal{T}^\pi\zeta_k^i(\boldsymbol{s}, \boldsymbol{a}))$.

In Algorithm 1, the gradient of actor and critic network, denoted as $\delta_{\theta^\mu}$ and $\delta_{\theta^Q}$, can be calculated as follows:

$$\begin{aligned}\delta_{\theta^\mu} &= (1/N) \sum \nabla_{\theta^\mu} \pi_{\theta^\mu}(\boldsymbol{s}_n)\mathbb{E}[\nabla_a Z_{\theta^Q}(\boldsymbol{s}_n, \boldsymbol{a})]_{a=\pi_{\theta^\mu}(\boldsymbol{s}_n)} \\ \delta_{\theta^Q} &= (1/N) \sum \nabla_{\theta^Q} \bar{d}_\infty(\Pi_{W_1}\mathcal{T}^\pi Z_{\theta^Q}(\boldsymbol{s}_n, \boldsymbol{a}_n), \Pi_{W_1}\mathcal{T}^\pi\widetilde{g}_i(\tau))\end{aligned} \tag{10}$$

where $\Pi_{W_1}$ represents a quantile approximation under the minimal 1-Wasserstein distance $W_1$.

# C ALGORITHM DETAILS AND PROOFS

## C.1 DETAILED DERIVATION OF THE OBJECTIVE FUNCTION

The aim of variational inference is to minimize the distance $D(q_\theta(\tau)||p(\tau|\mathcal{O})) = \mathbf{GSWD}_k(q_\theta(\tau), p(\tau|\mathcal{O}))$ between the variational distribution $q_\theta(\tau)$ and the posterior distribution $p(\tau|\mathcal{O})$. Let $\mathcal{P}_{trans} = p(\boldsymbol{s}, \boldsymbol{a}|\theta) p_D(\theta)$, $\tilde{x} = x/\mathcal{P}_{trans}$ and $\tilde{y} = y/\mathcal{P}_{trans}$ and recall Equation 3 in **Section 4.1**, i.e., the definition of $\mathbf{GSWD}_k$. Then the variational inference can be reformulated to the minimization problem:

$$\arg\min_{q_\theta(\tau)}\mathbf{GSWD}_k\left(q_\theta\left(\tau\right), p\left(\tau|\mathcal{O}\right)\right) = \arg\min_{q_\theta(\tau)}\mathbf{GSWD}_k\left(q\left(\boldsymbol{a}\right)\cdot\mathcal{P}_{trans}, p\left(\mathcal{O}|\tau\right)\cdot\mathcal{P}_{trans}\right)$$

$$= \arg\min_{q_\theta(\tau)}\left(\int_{\mathcal{X}_\theta} W_k^k\left(\mathcal{G}_{q(\boldsymbol{a})\cdot\mathcal{P}_{trans}}\left(\cdot,\widetilde{\theta}\right), \mathcal{G}_{p(\mathcal{O}|\tau)\cdot\mathcal{P}_{trans}}\left(\cdot,\widetilde{\theta}\right)\right)\mathrm{d}\widetilde{\theta}\right)^{\frac{1}{k}}$$

$$\overset{(i)}{=} \arg\min_{q_\theta(\tau)}\left(\int_{\mathcal{X}_\theta} W_k^k\left(\mathcal{P}_{trans}\cdot\mathcal{G}_{q(\boldsymbol{a})}\left(\cdot,\widetilde{\theta}\right), \mathcal{P}_{trans}\cdot\mathcal{G}_{p(\mathcal{O}|\tau)}\left(\cdot,\widetilde{\theta}\right)\right)\mathrm{d}\widetilde{\theta}\right)^{\frac{1}{k}}$$

$$= \arg\min_{q_\theta(\tau)}\inf_{\gamma\in\Gamma(\mathcal{P}_{trans}\cdot\mathcal{G}_{q(\boldsymbol{a})}, \mathcal{P}_{trans}\cdot\mathcal{G}_{p(\mathcal{O}|\tau)})}\left(\int_{\mathcal{X}_\theta}\int_{\mathcal{X}\times\mathcal{X}} d(x,y)^k\mathrm{d}\gamma(x,y)\mathrm{d}\widetilde{\theta}\right)^{\frac{1}{k}}$$

$$= \arg\min_{q_\theta(\tau)}\inf_{\gamma\in\Gamma(\mathcal{G}_{q(\boldsymbol{a})}, \mathcal{G}_{p(\mathcal{O}|\tau)})}\left(\int\int d(\mathcal{P}_{trans}\tilde{x}, \mathcal{P}_{trans}\tilde{y})^k\mathrm{d}\gamma(\mathcal{P}_{trans}\tilde{x}, \mathcal{P}_{trans}\tilde{y})\mathrm{d}\widetilde{\theta}\right)^{\frac{1}{k}}$$

$$\overset{(ii)}{=} \arg\min_{q_\theta(\tau)}\inf_{\gamma\in\Gamma(\mathcal{G}_{q(\boldsymbol{a})}, \mathcal{G}_{p(\mathcal{O}|\tau)})}\left(\int\int (\mathcal{P}_{trans}\cdot d(\tilde{x}, \tilde{y}))^k\mathrm{d}\gamma(\mathcal{P}_{trans}\tilde{x}, \mathcal{P}_{trans}\tilde{y})\mathrm{d}\widetilde{\theta}\right)^{\frac{1}{k}}$$

$$\overset{(iii)}{=} \arg\min_{q_\theta(\tau)}\mathcal{P}_{trans}\cdot\inf_{\gamma\in\Gamma(\mathcal{G}_{q(\boldsymbol{a})}, \mathcal{G}_{p(\mathcal{O}|\tau)})}\left(\int\int d(\tilde{x}, \tilde{y})^k\mathrm{d}\gamma(\tilde{x}, \tilde{y})\mathrm{d}\widetilde{\theta}\right)^{\frac{1}{k}}$$

$$= \arg\min_{q_\theta(\tau)}\mathcal{P}_{trans}\cdot\mathbf{GSWD}_k\left(q\left(\boldsymbol{a}\right), p\left(\mathcal{O}|\tau\right)\right)$$

(11)

where $(i)$ follows from the push-forward operator definition: $\mathcal{G}_\mu(l, \widetilde{\theta}) = \int_{\mathbb{G}^d}\delta(l - \langle x, \widetilde{\theta}\rangle)\mathrm{d}\mu$. $(ii)$ follows from $d(c\tilde{x}, c\tilde{y}) = cd(\tilde{x}, \tilde{y}), c\in(0,1)$ as $d$ is a metric. $(iii)$ follows from the fact that $\mathrm{d}\gamma(c\tilde{x}, c\tilde{y}) = \mathrm{d}\gamma(cx, cy) = \mathrm{d}\gamma(x, y) = \mathrm{d}\gamma(\tilde{x}, \tilde{y})$, since $\mathrm{d}\gamma(c\tilde{x}, c\tilde{y})$ is the measure of the subset of $\mathcal{X}\times\mathcal{X}$, which is just the re-scaled version of the subset $(\tilde{x}, \tilde{y})$ by the map $(x, y)\mapsto(x, y)$.

Equation 11 presents that the objective can be transformed to the minimization problem, i.e., $\arg\min_{q_\theta(\tau)}\mathbf{GSWD}_k\left(q\left(\boldsymbol{a}\right), p\left(\mathcal{O}|\tau\right)\right)$, where $p(\mathcal{O}|\tau)$ represents the optimality likelihood.

## C.2 DEFINITION OF DISTRIBUTIONAL TEMPORAL DIFFERENCE

We use distributional TD learning to estimate the distribution of state-action value, denoted as $Z$. In each iteration, we have the following:

$$\zeta_{k+1}^i(\boldsymbol{s}, \boldsymbol{a}) = \zeta_k^i(\boldsymbol{s}, \boldsymbol{a}) + l_{td}\boldsymbol{\Delta}_k^i$$
$$= \zeta_k^i(\boldsymbol{s}, \boldsymbol{a}) + l_{td}\times\bar{d}_\infty(\Pi_{W_1}\mathcal{T}^\pi(h_i(\boldsymbol{s}, \boldsymbol{a}, \boldsymbol{s}') + \gamma\zeta_k^i(\boldsymbol{s}')), \Pi_{W_1}\mathcal{T}^\pi\zeta_k^i(\boldsymbol{s}, \boldsymbol{a}))$$

(12)

where $\zeta_k^i\in S\times A$ represents the estimated distribution of the state-action distribution $Z$ in the $k$-th TD-learning-iteration for all $i = 0, ..., p$. The TD learning rate is denoted as $l_{td}$. The function $h_i : S\times A\times S\rightarrow\mathbb{R}$ maps the triple $(\boldsymbol{s}, \boldsymbol{a}, \boldsymbol{s}')$ to a *real number*. Specifically, $h_i$ is defined as $h_i = r$ when $i = 0$; and $h_i = g^i$ when $i\in[1, p]$. The distributional TD error $\boldsymbol{\Delta}_k^i$ in Equation 12 is calculated by $\bar{d}_\infty(\Pi_{W_1}\mathcal{T}^\pi(h_i(\boldsymbol{s}, \boldsymbol{a}, \boldsymbol{s}') + \gamma\zeta_k^i(\boldsymbol{s}')), \Pi_{W_1}\mathcal{T}^\pi\zeta_k^i(\boldsymbol{s}, \boldsymbol{a}))$.

## C.3 PROOFS

Here we give the proofs of **Proposition 1** *(Pseudo-metric)*, **Proposition 2** *(Policy Evaluation)*, **Proposition 3** *(Policy Improvement)*, **Theorem 1** *(Global Convergence)* and **Theorem 2** *(Global Convergence Rate)* in **Section 5**;

**Proposition 1.** *(Pseudo-metric)*: Given two probability measures $\mu, \nu \in P_k(\mathcal{X})$, and a mapping $g_{rl} : \mathcal{X} \rightarrow \mathcal{R}_{\widetilde{\theta}}$, the adaptive generalized sliced Wasserstein distance A-GSWD of order $k \in [1, \infty)$ is a pseudo-metric that satisfies non-negativity, symmetry, the triangle inequality and $\mathbf{A} - \mathbf{GSWD}_k(\mu, \mu) = 0$.

*Proof*: The non-negativity property naturally arises from the fact that the Wasserstein distance $W_k$ is a metric Villani et al. (2009). To prove symmetry, since the k-Wasserstein distance is a metric Villani et al. (2009):

$$W_k\left(\mathcal{G}_\mu(\cdot, \widetilde{\theta}; g_{rl}), \mathcal{G}_\nu(\cdot, \widetilde{\theta}; g_{rl})\right) = W_k\left(\mathcal{G}_\nu(\cdot, \widetilde{\theta}; g_{rl}), \mathcal{G}_\mu(\cdot, \widetilde{\theta}; g_{rl})\right)$$

Thus, there exists Chen et al. (2020):

$$\mathbf{A} - \mathbf{GSWD_k}(\mu, \nu) = \left(\int_{\mathcal{R}_{\widetilde{\theta}}} W_k^k\left(\mathcal{G}_\mu(\cdot, \widetilde{\theta}; g_{rl}), \mathcal{G}_\nu(\cdot, \widetilde{\theta}; g_{rl})\right) \mathrm{d}\widetilde{\theta}\right)^{\frac{1}{k}}$$

$$= \left(\int_{\mathcal{R}_{\widetilde{\theta}}} W_k^k\left(\mathcal{G}_\nu(\cdot, \widetilde{\theta}; g_{rl}), \mathcal{G}_\mu(\cdot, \widetilde{\theta}; g_{rl})\right) \mathrm{d}\widetilde{\theta}\right)^{\frac{1}{k}} = \mathbf{A} - \mathbf{GSWD_k}(\nu, \mu)$$

Therefore, symmetry holds. Then, we prove the triangle inequality. Since the triangle inequality holds for the Wasserstein distance, we can obtain $W_k\left(\mathcal{G}_{\mu_1}, \mathcal{G}_{\mu_3}\right) \leq W_k\left(\mathcal{G}_{\mu_1}, \mathcal{G}_{\mu_2}\right) + W_k\left(\mathcal{G}_{\mu_2}, \mathcal{G}_{\mu_3}\right)$. Thus, there exists:

$$\mathbf{A} - \mathbf{GSWD}_k(\mu_1, \mu_3) = \left(\int_{\mathcal{R}_{\widetilde{\theta}}} W_k^k\left(\mathcal{G}_{\mu_1}, \mathcal{G}_{\mu_3}\right) \mathrm{d}\widetilde{\theta}\right)^{\frac{1}{k}}$$

$$\leq \left(\int_{\mathcal{R}_{\widetilde{\theta}}} W_k^k\left(\mathcal{G}_{\mu_1}, \mathcal{G}_{\mu_2}\right) + W_k^k\left(\mathcal{G}_{\mu_2}, \mathcal{G}_{\mu_3}\right) \mathrm{d}\widetilde{\theta}\right)^{\frac{1}{k}} \tag{13}$$

$$\leq \left(\int_{\mathcal{R}_{\widetilde{\theta}}} W_k^k\left(\mathcal{G}_{\mu_1}, \mathcal{G}_{\mu_2}\right) \mathrm{d}\widetilde{\theta}\right)^{\frac{1}{k}} + \left(\int_{\mathcal{R}_{\widetilde{\theta}}} W_k^k\left(\mathcal{G}_{\mu_2}, \mathcal{G}_{\mu_3}\right) \mathrm{d}\widetilde{\theta}\right)^{\frac{1}{k}}$$

where the derivation of Equation 13 is based on the Minkowski inequality Bahouri et al. (2011), which establishes that $\mathbf{A} - \mathbf{GSWD}_k$ satisfies the triangle inequality.

For the identity of indiscernibles, we can firstly obtain $W_k(\mathcal{G}_{\mu_1}, \mathcal{G}_{\mu_2}) = 0$ if and only if $\mathcal{G}_{\mu_1} = \mathcal{G}_{\mu_2}$. Thus there exists $\mathbf{A} - \mathbf{GSWD}_k(\mathcal{G}_{\mu_1}, \mathcal{G}_{\mu_2}) = 0$ if and only if $\mathcal{G}_{\mu_1} = \mathcal{G}_{\mu_2}$, i.e., $g_{rl}$ in Equation 5 is an injective mapping. ∎

Based on the preceding proof, we can deduce the condition in **Remark 1** that under which the A-GSWD of order $k \in [1, \infty)$ exhibits the properties of a pseudo-metric.

**Proposition 2.** *(Policy Evaluation) Dabney et al. (2018); Wang et al. (2023))*: we consider a quantile approximation $\Pi_{W_1}$ under the minimal 1-Wasserstein distance $W_1$, the *Bellman operator* $\mathcal{T}^\pi$ under a deterministic policy $\pi$ and $Z_{k+1}(\boldsymbol{s}, \boldsymbol{a}) = \Pi_{W_1} \mathcal{T}^\pi Z_k(\boldsymbol{s}, \boldsymbol{a})$. The sequence $Z_k(\boldsymbol{s}, \boldsymbol{a})$ converges to a unique fixed point $\widetilde{Z}_\pi$ under the maximal form of $\infty$-Wasserstein metric $\bar{d}_\infty$.

*Proof*: We recall a contraction proved in Dabney et al. (2018) over the Wasserstein Metric:

$$\bar{d}_\infty(\Pi_{W_1} \mathcal{T}^\pi Z_1, \Pi_{W_1} \mathcal{T}^\pi Z_2) \leq \bar{d}_\infty(Z_1, Z_2) \tag{14}$$

where Equation 14 implies that the combined operator $\Pi_{W_1} \mathcal{T}^\pi$ is an $\infty$-contraction. Based on Banach's fixed point theorem, $\mathcal{T}^\pi$ has a unique fixed point, i.e., $\widetilde{Z}_\pi$. Furthermore, the definition of *Bellman optimality operator*, defined as Equation 8, which implies that all moments of $Z$ are bounded. Therefore, we conclude that the sequence $Z_k(\boldsymbol{s}, \boldsymbol{a})$ converges to $\widetilde{Z}_\pi$ in $\bar{d}_\infty$ for $p \in [1, \infty]$. ∎

**Proposition 3.** *(Policy Improvement)*: Given an old policy $\boldsymbol{\pi_{old}}$, a new policy $\boldsymbol{\pi_{new}}$ and $Q(s, a) = \mathbb{E}[Z(s, a)]$, there exists $Q^{\boldsymbol{\pi_{new}}}(s, a) \geq Q^{\boldsymbol{\pi_{old}}}(s, a)$ when performing Algorithm 1, $\forall s \in \mathcal{S}$ and $\forall a \in \mathcal{A}$ *if and only if* the reward operator family $\mathcal{F} = \{\mathcal{F}_r, \mathcal{F}_g\}$ satisfies the both **Conditions**.

**Conditions.** Regarding $\mathcal{F}$, the reward operator family: (*i*) $\mathcal{F}_r$ is monotonically increasing and continuously defined on $(0, 1]$, and the range covers $[r_{\min}, r_{\max}]$; and (*ii*) $\mathcal{F}_g$ is monotonically decreasing and continuously defined on $(0, 1]$, and the range covers $[r_{\min}, r_{\max}]$.

*Proof*: We recall that $\{\mathcal{F}_r, \mathcal{F}_g\}$ are two operators defined as $\widetilde{r}(\tau) := \mathcal{F}_r \cdot p(\mathcal{O}_r | \tau)$ and $\widetilde{g}^i(\tau) := \mathcal{F}_g \cdot p(\mathcal{O}_g | \tau)$, respectively. Since the two optimization objectives in policy updating, i.e., $\max \mathbb{E}[\mathcal{F}_r \cdot p(\mathcal{O}_r | \tau)]$ and $\min \mathbb{E}[\mathcal{F}_g \cdot p(\mathcal{O}_g | \tau)]$ (see Equation 4 in **Section 4**), and $p(\mathcal{O} | \tau)$ is defined on $(0, 1]$, we can conclude the both **Conditions** that (*i*) $\mathcal{F}_r$ is monotonically increasing and continuously defined on $(0, 1]$, and the range covers $[r_{\min}, r_{\max}]$; (*ii*) $\mathcal{F}_g$ is monotonically decreasing and continuously defined on $(0, 1]$, and the range covers $[r_{\min}, r_{\max}]$.

Then based on Equation 8, there exists:

$$
\begin{aligned}
V^\pi(s_t) = \mathbb{E}[Q(s_t, \pi(s_t))] &\leq \max_{a' \in \mathcal{A}} \mathbb{E}[Q(s_t, a')] \\
&= \mathbb{E}[Q(s_t, \pi'(s_t))]
\end{aligned}
\tag{15}
$$

where $\mathbb{E}_\pi[\cdot] = \sum_{a \in A} \pi(a|s)[\cdot]$, and $V^\pi(s) = \mathbb{E}_\pi \mathbb{E}[Z_k(s, a)]$ is the value function. According to Equation 15 and Equation 8, it yields:

$$
\begin{aligned}
Q^{\boldsymbol{\pi_{old}}} &= Q^{\boldsymbol{\pi_{old}}}(s_t, \boldsymbol{\pi_{old}}(s_t)) \\
&= r_{t+1} + \gamma \mathbb{E}_{s_{t+1}} \mathbb{E}_{\boldsymbol{\pi_{old}}} Q^{\boldsymbol{\pi_{old}}}(s_{t+1}, \boldsymbol{\pi_{old}}(s_{t+1})) \\
&\overset{(i)}{\leq} r_{t+1} + \gamma \mathbb{E}_{s_{t+1}} \mathbb{E}_{\boldsymbol{\pi_{new}}} Q^{\boldsymbol{\pi_{old}}}(s_{t+1}, \boldsymbol{\pi_{new}}(s_{t+1})) \\
&\leq r_{t+1} + \mathbb{E}_{s_{t+1}} \mathbb{E}_{\boldsymbol{\pi_{new}}}[\gamma r_{t+2} \\
&\quad + \gamma^2 \mathbb{E}_{s_{t+2}} Q^{\boldsymbol{\pi_{old}}}(s_{t+2}, \boldsymbol{\pi_{new}}(s_{t+2}))|] \\
&\leq r_{t+1} + \mathbb{E}_{s_{t+1}} \mathbb{E}_{\boldsymbol{\pi_{new}}}[\gamma r_{t+2} + \gamma^2 r_{t+3} + ...] \\
&= r_{t+1} + \mathbb{E}_{s_{t+1}} V^{\boldsymbol{\pi_{new}}}(s_{t+1}) \\
&= Q^{\boldsymbol{\pi_{new}}}
\end{aligned}
\tag{16}
$$

where $(i)$ relies on Equation 15, and $\boldsymbol{\pi_{new}}$ corresponds to the maximum $Q$ in the Bellman function. Therefore, we have $Q^{\boldsymbol{\pi_{new}}}(s, a) \geq Q^{\boldsymbol{\pi_{old}}}(s, a)$ ∎

Then we provide **Lemma 1** and the proof of **Theorem 1**.

**Lemma 1.** (Bellemare et al. (2017)): The *Bellman operator* $\mathcal{T}^\pi$ is a $p$-contraction under the $p$-Wasserstein metric $\overline{d}_p$.

**Theorem 1.** *(Global Convergence)*: Given the policy in the $i$-th policy improvement $\boldsymbol{\pi^i}$, $\boldsymbol{\pi^i} \to \boldsymbol{\pi^*}$ and $i \to \infty$, there exists $Q^{\boldsymbol{\pi^*}}(s, a) \geq Q^{\boldsymbol{\pi^i}}(s, a)$ *if and only if* the reward operator family $\mathcal{F}$ satisfies the both **Conditions**.

*Proof*: Since **Proposition 3** suggests $Q^{\boldsymbol{\pi_{i+1}}}(s, a) \geq Q^{\boldsymbol{\pi_i}}(s, a)$, the sequence $Q^{\boldsymbol{\pi_i}}(s, a)$ is monotonically increasing *if and only if* the reward operator family $\mathcal{F}$ satisfies the both **Conditions**. Furthermore, **Lemma 1** implies that the the state-action distribution $Z$ over $\mathbb{R}$ has bounded $p$-th moment, so the first moment of $Z$, i.e., $Q^{\boldsymbol{\pi_i}}(s, a)$, is upper bounded. Therefore, the sequence $Q^{\boldsymbol{\pi_i}}(s, a)$ converges to an upper limit $Q^{\boldsymbol{\pi_*}}(s, a)$ with $\forall s \in \mathcal{S}$ and $\forall a \in \mathcal{A}$. ∎

To prove **Theorem 2**, we provide **Lemma 2** and its proof below.

**Lemma 2.** *(Convergence rate of neural TD learning)*: Let $m$ be the width of the actor-critic networks, and $\bar{Z}_t = \frac{1}{N} \sum_{i=1}^{N} \delta_{q_i(\boldsymbol{s}, \boldsymbol{a})}$ be an estimator of $Z_t^i$. In the TD learning, with probability at least $1 - \delta$, there exists

$$
\begin{aligned}
\left\| \Pi_{W_1} \bar{Z}_t - \Pi_{W_1} Z_t^* \right\| &\leq \Theta(m^{-\frac{H}{4}}) \\
&+ \Theta([(1-\gamma)K]^{-\frac{1}{2}}[1 + \log^{\frac{1}{2}} \delta^{-1}])
\end{aligned}
\tag{17}
$$

*Proof*: Based on Gluing lemma of Wasserstein distance $W_p$ Villani (2009); Clement & Desch (2008), there exists:

$$
\left\| \Pi_{W_1}\bar{Z}_t - \Pi_{W_1}Z_t^* \right\| \overset{(i)}{=} \sum_{i=1}^{N} \left\| \bar{q}_t^i - q_t^{i,*} \right\|
$$

$$
= \sum_{i=1}^{N} \left\| f_i^{(H)}((\boldsymbol{s},\boldsymbol{a}), \theta_{K_{td}}^Q) - f_i^{(H)}((\boldsymbol{s},\boldsymbol{a}), \theta^{Q^*}) \right\|
$$

$$
\leq \sum_{i=1}^{N} \left\| f_i^{(H)}((\boldsymbol{s},\boldsymbol{a}), \theta_{K_{td}}^Q) - f_{0,i}^{(H)}((\boldsymbol{s},\boldsymbol{a}), \theta^Q) \right\|
$$

$$
+ \sum_{i=1}^{N} \left\| f_{0,i}^{(H)}((\boldsymbol{s},\boldsymbol{a}), \theta_{K_{td}}^Q) - f_i^{(H)}((\boldsymbol{s},\boldsymbol{a}), \theta^{Q^*}) \right\|
$$

$$
\overset{(ii)}{\leq} \Theta(m^{-\frac{H}{4}}) + \sum_{i=1}^{N} \left\| f_{0,i}^{(H)}((\boldsymbol{s},\boldsymbol{a}), \theta_{K_{td}}^Q) - f_i^{(H)}((\boldsymbol{s},\boldsymbol{a}), \theta^{Q^*}) \right\|
$$

$$
\overset{(iii)}{\leq} \Theta(m^{-\frac{H}{4}}) + \Theta([(1-\gamma)K_{td}]^{-\frac{1}{2}}[1 + \log^{\frac{1}{2}}\delta^{-1}])
$$

(18)

where $H$ denotes the layers of the neural network. (i) holds relying on 1-Wasserstein distance $W_1$ and one-dimensional quantile $q_t^{i,*}$. As each quantile can be viewed as a form of local linearization, (ii) follows from *Lemma 5.1* in Cai et al. (2019):

$$
\sum_{i=1}^{N} \left\| f_i^{(H)}((\boldsymbol{s},\boldsymbol{a}), \theta_{K_{td}}^Q) - f_{0,i}^{(H)}((\boldsymbol{s},\boldsymbol{a}), \theta^Q) \right\|^2 \leq \frac{1}{m^H} \sum_{i=1}^{N} b_r
$$

$$
\left| [(\mathbf{1}(W_i^{(h)} x_i^{(h-1)} > 0) - \mathbf{1}(W_i^{(0)} x_i^{(h-1)} > 0)) \cdot W_i^{(h)} x_i^{(h-1)}]^2 \right|
$$

$$
\leq \frac{4C_0}{m^H} \sum_{i=1}^{N} \sum_{r=1}^{m} \mathbf{1}(\left| W_{i,r}^{(0)} x_i^{(h-1)} \right| \leq \left\| W_{i,r}^{(h)} - W_{i,r}^{(0)} \right\|_2)]
$$

$$
\leq \frac{4C_0}{m^H} (\sum_{r=1}^{m} \left\| W_{i,r}^{(h)} - W_{i,r}^{(0)} \right\|_2^2)^{\frac{1}{2}} (\sum_{r=1}^{m} \left\| \frac{1}{W_{i,r}^{(0)}} \right\|_2^2)^{\frac{1}{2}} \leq \frac{4C_0 C_1}{m^{\frac{H}{2}}}
$$

(19)

where the constant $C_0 > 0$ and $C_1 > 0$. Thus we upper bound $\sum_{i=1}^{N} \left\| f_i^{(H)} - f_{0,i}^{(H)} \right\| \leq \Theta(m^{-\frac{H}{4}})$, which holds (i) in Equation 18. Then (ii) follows from *Lemma 1* in Rahimi & Recht (2008), with probability at least $1 - \delta$, there exists:

$$
\sum_{i=1}^{N} \left\| f_{0,i}^{(H)}((\boldsymbol{s},\boldsymbol{a}), \theta_{K_{td}}^Q) - f_i^{(H)}((\boldsymbol{s},\boldsymbol{a}), \theta^{Q^*}) \right\|
$$

$$
\leq \frac{1}{\sqrt{1-\gamma}} \sum_{i=1}^{N} \left\| f_{0,i}^{(H)}((\boldsymbol{s},\boldsymbol{a}), \theta_{K_{td}}^{Q^\pi}) - f_i^{(H)}((\boldsymbol{s},\boldsymbol{a}), \theta^{Q^*}) \right\|
$$

$$
\leq \frac{C_3}{\sqrt{(1-\gamma)K_{td}}} (1 + \sqrt{\log \frac{1}{\delta}})
$$

(20)

where (iii) holds, and therefore Equation 18 holds. ■

**Theorem 2.** *(Global Convergence Rate)*: Let $m$ and $H$ be the width and the layer of neural network, $K_{td} = (1-\gamma)^{-\frac{3}{2}} m^{\frac{H}{2}}$ be the iterations required for convergence of the distributional TD learning (defined in Equation 12), $l_Q = \frac{1}{\sqrt{T}}$ be the policy update (in *Line 4* of Algorithm 1) and $\tau_c = \Theta(\frac{1}{(1-\gamma)\sqrt{T}}) + \Theta(\frac{1}{(1-\gamma)Tm^{\frac{H}{4}}})$ be the tolerance (in *Line 3* of Algorithm 1). There exists a global convergence rate of $\Theta(1/\sqrt{T})$, and a sublinear rate of $\Theta(1/\sqrt{T})$ if the constraints are violated with

an error of $\Theta(1/m^{\frac{H}{4}})$, with probability at least $1 - \delta$. Importantly, this conclusion holds if and only if the reward operator family $\mathcal{F}$ satisfies both **Conditions**.

*Proof*: **Proposition 3** suggests that the sequence $Q^{\pi_i}(s, a)$ achieves global convergence *if and only if* the reward operator family $\mathcal{F}$ satisfies **Conditions**. Then we let $\triangle_{\theta Q} = \theta_{t+1}^Q - \theta_t^Q$, and suppose the critic networks are $H$-layer neural networks. Based on *Lemma 6.1* in Kakade & Langford (2002), there exists

$$(1 - \gamma)[\mathcal{J}_r(\boldsymbol{\pi}^*) - \mathcal{J}_r(\boldsymbol{\pi}_t)]$$

$$= \mathbb{E}[Q_{\pi_t}(\boldsymbol{s}, \boldsymbol{a}) - \mathbb{E}Q_{\pi_t}(\boldsymbol{s}, \boldsymbol{a}')]$$

$$= \mathbb{E}[\nabla_\theta f^{(H)}((\boldsymbol{s}, \boldsymbol{a}), \theta^Q)^{\mathrm{T}} - \mathbb{E}[\nabla_\theta f^{(H)}((\boldsymbol{s}, \boldsymbol{a}'), \theta^Q)^{\mathrm{T}}]]\triangle_{\theta Q}$$

$$+ \mathbb{E}[Q_{\pi_t}(\boldsymbol{s}, \boldsymbol{a}) - \nabla_\theta f^{(H)}((\boldsymbol{s}, \boldsymbol{a}), \theta^Q)^{\mathrm{T}}\triangle_{\theta Q}]$$

$$+ \mathbb{E}[\nabla_\theta f^{(H)}((\boldsymbol{s}, \boldsymbol{a}'), \theta^Q)^{\mathrm{T}}\triangle_{\theta Q} - Q_{\pi_t}(\boldsymbol{s}, \boldsymbol{a}')]$$

$$= \frac{1}{l_Q}\Big[l_Q\mathbb{E}[\nabla_\theta \log(\boldsymbol{\pi}_t(\boldsymbol{a}|\boldsymbol{s}))^{\mathrm{T}}]\triangle_{\theta Q} - \frac{l_Q^2 \mathcal{L}_f}{2}\|\triangle_{\theta Q}\|_2^2\Big]$$

$$+ \mathbb{E}[Q_{\pi_t}(\boldsymbol{s}, \boldsymbol{a}) - \nabla_\theta f^{(H)}((\boldsymbol{s}, \boldsymbol{a}), \theta^Q)^{\mathrm{T}}\triangle_{\theta Q}] + \frac{l_Q \mathcal{L}_f}{2}\|\triangle_{\theta Q}\|_2^2$$

$$+ \mathbb{E}[\nabla_\theta f^{(H)}((\boldsymbol{s}, \boldsymbol{a}'), \theta^Q)^{\mathrm{T}}\triangle_{\theta Q} - Q_{\pi_t}(\boldsymbol{s}, \boldsymbol{a}')]$$

$$\overset{(i)}{\leq} \frac{1}{l_Q}\mathbb{E}[\log(\frac{\boldsymbol{\pi}_{t+1}(\boldsymbol{a}|\boldsymbol{s})}{\boldsymbol{\pi}_t(\boldsymbol{a}|\boldsymbol{s})})] + \frac{l_Q \mathcal{L}_f}{2}\|\triangle_{\theta Q}\|_2^2 \qquad (21)$$

$$+ \sqrt{\mathbb{E}[Q_{\pi_t}(\boldsymbol{s}, \boldsymbol{a}) - f^{(H)}((\boldsymbol{s}, \boldsymbol{a}), \triangle_{\theta Q})]^2}$$

$$+ \sqrt{\mathbb{E}[f^{(H)}((\boldsymbol{s}, \boldsymbol{a}), \triangle_{\theta Q}) - \nabla_\theta f^{(H)}((\boldsymbol{s}, \boldsymbol{a}), \theta^Q)^{\mathrm{T}}\triangle_{\theta Q}]^2}$$

$$+ \sqrt{\mathbb{E}[\nabla_\theta f^{(H)}((\boldsymbol{s}, \boldsymbol{a}'), \theta^Q)^{\mathrm{T}}\triangle_{\theta Q} - f^{(H)}((\boldsymbol{s}, \boldsymbol{a}'), \triangle_{\theta Q})]^2}$$

$$+ \sqrt{\mathbb{E}[f^{(H)}((\boldsymbol{s}, \boldsymbol{a}'), \triangle_{\theta Q}) - Q_{\pi_t}(\boldsymbol{s}, \boldsymbol{a}')]^2}$$

$$= \frac{1}{l_Q}\big[\mathbb{E}[\mathcal{D}_{KL}(\boldsymbol{\pi}^*\|\boldsymbol{\pi}_t)] - \mathbb{E}[\mathcal{D}_{KL}(\boldsymbol{\pi}^*\|\boldsymbol{\pi}_{t+1})]\big]$$

$$+ 2\sqrt{\mathbb{E}[f^{(H)}((\boldsymbol{s}, \boldsymbol{a}), \triangle_{\theta Q}) - \nabla_\theta f^{(H)}((\boldsymbol{s}, \boldsymbol{a}), \theta^Q)^{\mathrm{T}}\triangle_{\theta Q}]^2}$$

$$+ 2\sqrt{\mathbb{E}[Q_{\pi_t}(\boldsymbol{s}, \boldsymbol{a}) - f^{(H)}((\boldsymbol{s}, \boldsymbol{a}), \triangle_{\theta Q})]^2} + \frac{l_Q \mathcal{L}_f}{2}\|\triangle_{\theta Q}\|_2^2$$

where (i) follows from the $\mathcal{L}_f$-Lipschitz property of $\log(\boldsymbol{\pi}_t(\boldsymbol{a}|\boldsymbol{s}))$. Next, we upper bound the term $\sqrt{\mathbb{E}[f^{(H)}((\boldsymbol{s}, \boldsymbol{a}), \triangle_{\theta Q}) - \nabla_\theta f^{(H)}((\boldsymbol{s}, \boldsymbol{a}), \theta^Q)^{\mathrm{T}}\triangle_{\theta Q}]^2}$ as shown below.

$$\sqrt{\mathbb{E}[f^{(H)}((\boldsymbol{s}, \boldsymbol{a}), \triangle_{\theta Q}) - \nabla_\theta f^{(H)}((\boldsymbol{s}, \boldsymbol{a}), \theta^Q)^{\mathrm{T}}\triangle_{\theta Q}]^2}$$

$$= \sum_{i=1}^N \left\|f_i^{(H)}((\boldsymbol{s}, \boldsymbol{a}), \triangle_{\theta Q}) - \nabla_\theta f_i^{(H)}((\boldsymbol{s}, \boldsymbol{a}), \theta^Q)^{\mathrm{T}}\triangle_{\theta Q}\right\|$$

$$\leq \sum_{i=1}^N \big[\left\|f_i^{(H)}((\boldsymbol{s}, \boldsymbol{a}), \triangle_{\theta Q}) - \nabla_\theta f_{0,i}^{(H)}((\boldsymbol{s}, \boldsymbol{a}), \theta^Q)^{\mathrm{T}}\triangle_{\theta Q}\right\| \qquad (22)$$

$$+ \left\|\nabla_\theta f_{0,i}^{(H)}((\boldsymbol{s}, \boldsymbol{a}), \theta^Q)^{\mathrm{T}}\triangle_{\theta Q} - \nabla_\theta f_i^{(H)}((\boldsymbol{s}, \boldsymbol{a}), \theta^Q)^{\mathrm{T}}\triangle_{\theta Q}\right\|\big]$$

$$= 2\sum_{i=1}^N \left\|f_i^{(H)}((\boldsymbol{s}, \boldsymbol{a}), \triangle_{\theta Q}) - f_{0,i}^{(H)}((\boldsymbol{s}, \boldsymbol{a}), \triangle_{\theta Q})\right\|$$

$$\overset{(ii)}{\leq} \frac{4\sqrt{C_0 C_1}}{m^{\frac{H}{4}}}$$

where (ii) follows from Equation 19. Then, in order to upper bound $\sqrt{\mathbb{E}[Q_{\pi_t}(\boldsymbol{s}, \boldsymbol{a}) - f^{(H)}((\boldsymbol{s}, \boldsymbol{a}), \triangle_{\theta Q})]^2}$, taking expectation of Equation 21 from $t = 0$ to

$T - 1$, yields

$$
\begin{aligned}
&(1 - \gamma)\big[\mathcal{J}_r(\boldsymbol{\pi}^*) - \mathbb{E}[\mathcal{J}_r(\boldsymbol{\pi})]\big] \\
&= (1 - \gamma)\frac{1}{T}\sum_{t=0}^{T-1}[\mathcal{J}_r(\boldsymbol{\pi}^*) - \mathcal{J}_r(\boldsymbol{\pi}_t)] \\
&\leq \frac{1}{T}\Big[\frac{1}{l_Q}\mathbb{E}[\mathcal{D}_{KL}(\boldsymbol{\pi}^*\|\boldsymbol{\pi}_t)] + \frac{8T\sqrt{C_0 C_1}}{m^{\frac{H}{4}}} + \frac{Tl_Q\mathcal{L}_f}{2}d_\theta^2 \\
&\quad + 2\sum_{t=0}^{T-1}\sum_{i=1}^{N}\Big\| f_i^{(H)}((\boldsymbol{s},\boldsymbol{a}),\theta_{t+1}^Q - \theta_t^Q) - f_i^{(H)}((\boldsymbol{s},\boldsymbol{a}),\theta^{Q^*})\Big\|\Big] \\
&= \frac{\mathbb{E}[\mathcal{D}_{KL}(\boldsymbol{\pi}^*\|\boldsymbol{\pi}_t)]}{l_Q T} + \frac{8\sqrt{C_0 C_1}}{m^{\frac{H}{4}}} + \frac{l_Q\mathcal{L}_f}{2}d_\theta^2 \\
&\quad + \frac{2}{T}\sum_{i=1}^{N}\Big\| f_i^{(H)}((\boldsymbol{s},\boldsymbol{a}),\theta_{K_{td},t}^Q) - f_i^{(H)}((\boldsymbol{s},\boldsymbol{a}),\theta^{Q^*})\Big\| \\
&\overset{(iii)}{\leq} \frac{\mathbb{E}[\mathcal{D}_{KL}(\boldsymbol{\pi}^*\|\boldsymbol{\pi}_t)]}{l_Q T} + \frac{8\sqrt{C_0 C_1}}{m^{\frac{H}{4}}} + \frac{l_Q\mathcal{L}_f}{2}d_\theta^2 \\
&\quad + \frac{4\sqrt{C_0 C_1}}{Tm^{\frac{H}{4}}} + \frac{2C_3}{T\sqrt{(1-\gamma)K_{td}}}(1 + \sqrt{\log\frac{1}{\delta}})
\end{aligned}
\tag{23}
$$

where (iii) follows from **Lemma 2** (Equation 18). Thus, substituting $K_{td} = (1-\gamma)^{-1}m^{\frac{H}{2}}$ and $l_Q = \Theta(1/\sqrt{T})$ into Equation 23, with probability at least $1 - \delta$, yields:

$$
\begin{aligned}
\mathcal{J}_r(\boldsymbol{\pi}^*) - \mathbb{E}[\mathcal{J}_r(\boldsymbol{\pi})] &\leq C_5\frac{1}{(1-\gamma)\sqrt{T}} + C_6\frac{1}{(1-\gamma)m^{\frac{H}{4}}} \\
&\quad + C_7\frac{1}{(1-\gamma)Tm^{\frac{H}{4}}} + 2C_3\frac{\sqrt{\log\frac{1}{\delta}}}{(1-\gamma)Tm^{\frac{H}{4}}} \\
&\leq \Theta\big(\frac{1}{(1-\gamma)\sqrt{T}}\big) + \Theta\big(\frac{1}{(1-\gamma)Tm^{\frac{H}{4}}}\sqrt{\log\frac{1}{\delta}}\big)
\end{aligned}
\tag{24}
$$

where $C_5 = \mathbb{E}[\mathcal{D}_{KL}(\boldsymbol{\pi}^*\|\boldsymbol{\pi}_t)] + \frac{\mathcal{L}_f d_\theta^2}{2}$, $C_6 = 8\sqrt{C_0 C_1}$ and $C_7 = 4\sqrt{C_0 C_1} + 2C_3$. Therefore, there exists:

$$
\begin{aligned}
\mathcal{J}_r(\boldsymbol{\pi}^*) - \mathbb{E}[\mathcal{J}_r(\boldsymbol{\pi})] &\leq \Theta\big(\frac{1}{(1-\gamma)\sqrt{T}}\big) \\
&\quad + \Theta\big(\frac{1}{(1-\gamma)Tm^{\frac{H}{4}}}\sqrt{\log\frac{1}{\delta}}\big)
\end{aligned}
\tag{25}
$$

where Equation 25 suggests that there exists a global convergence rate of $\Theta(1/\sqrt{T})$, with probability at least $1 - \delta$.

Following *Line 6* in Algorithm 1 and recalling Equation 21, Equation 22 and Equation 23, the convergence process is similarly stated for the constraint approximation $\mathcal{J}_g^i(\boldsymbol{\pi})$, $\forall i \in [1, p]$ here

$$
\begin{aligned}
\mathbb{E}[\mathcal{J}_g^i(\boldsymbol{\pi})] - \mathcal{J}_g^i(\boldsymbol{\pi}^*) &\leq \Theta\big(\frac{1}{(1-\gamma)\sqrt{T}}\big) \\
&\quad + \Theta\big(\frac{1}{(1-\gamma)Tm^{\frac{H}{4}}}\sqrt{\log\frac{1}{\delta}}\big)
\end{aligned}
\tag{26}
$$

the constraint violation is then bounded below

$$
\begin{aligned}
\mathbb{E}[\mathcal{J}_g^i(\boldsymbol{\pi})] - \boldsymbol{b}_i &\leq \big[\mathcal{J}_g^i(\boldsymbol{\pi}^*) - \boldsymbol{b}_i\big] + \big[\mathbb{E}[\mathcal{J}_g^i(\boldsymbol{\pi})] - \mathcal{J}_g^i(\boldsymbol{\pi}^*)\big] \\
&\leq \boldsymbol{\tau}_c + \big[\mathbb{E}[\mathcal{J}_g^i(\boldsymbol{\pi})] - \mathcal{J}_g^i(\boldsymbol{\pi}^*)\big] \\
&\leq \boldsymbol{\tau}_c + \Theta\big(\frac{1}{(1-\gamma)\sqrt{T}}\big) + \Theta\big(\frac{1}{(1-\gamma)Tm^{\frac{H}{4}}}\sqrt{\log\frac{1}{\delta}}\big)
\end{aligned}
\tag{27}
$$

where we have $\boldsymbol{\tau}_c = \Theta(\frac{1}{(1-\gamma)\sqrt{T}}) + \Theta(\frac{1}{(1-\gamma)Tm^{\frac{H}{4}}})$, therefore, we obtain:

$$
\begin{aligned}
\mathbb{E}[\mathcal{J}_g^i(\boldsymbol{\pi})] - \boldsymbol{b}_i &\leq \Theta(\frac{1}{(1-\gamma)\sqrt{T}}) \\
&\quad + \Theta(\frac{1}{(1-\gamma)Tm^{\frac{H}{4}}}\sqrt{\log\frac{1}{\delta}})
\end{aligned}
\tag{28}
$$

where Equation 28 suggests that there exists a sublinear rate of $\Theta(1/\sqrt{T})$ if the constraints are violated with an error of $\Theta(1/m^{\frac{H}{4}})$, with probability at least $1 - \delta$. ∎

# D    EXPERIMENT SUPPLEMENTARY

## D.1    EXPERIMENTAL SETTING

The parameter setting of AWaVO is shown in Table 1, and the technical specification of the quadrotor is shown in Table 2.

Table 1: Parameter Setting of AWaVO

| Parameters | Definition | Values |
|---|---|---|
| $l_{\mu,cart}$ | Learning rate of actor in Cartpole Xu et al. (2021) | 0.0005 |
| $l_{\theta,cart}$ | Learning rate of critic in Cartpole Xu et al. (2021) | 0.0005 |
| $l_{\mu,acro}$ | Learning rate of actor in Acrobot Xu et al. (2021) | 0.005 |
| $l_{\theta,acro}$ | Learning rate of critic in Acrobot Xu et al. (2021) | 0.005 |
| $l_{\mu,guard}$ | Learning rate of actor in Walker and Drone Zhao et al. (2023) | 0.001 |
| $l_{\theta,guard}$ | Learning rate of critic in Walker and Drone Zhao et al. (2023) | 0.001 |
| $\mu$ | Actor neural network: fully connected with $H$ hidden layers ($m$ neurons per hidden layer) | - |
| $\theta$ | Critic neural network: fully connected with $H$ hidden layers ($m$ neurons per hidden layer) | - |
| $D$ | Replay memory capacity | $10^6$ |
| $B$ | Batch size | 128 |
| $\gamma$ | Discount rate | 0.998 |
| - | Training episodes | 1000 |
| $m$ | the width of neural network | 128 |
| $H$ | the layer of neural network | 2 |
| $T$ | Length in each episode | 500 |
| $N$ | Time steps | 20 |

Table 2: Technical Specification of Hardware

| No. | Component | Specific Model |
|---|---|---|
| 1 | Frame | QAV250 |
| 2 | Sensor - Depth Camera | Intel RealSense D435i |
| 3 | Sensor - Down-view Rangefinder | Holybro ST VL53L1X |
| 4 | Flight Controller | Pixhawk 4 |
| 5 | Motors | T-Motor F60 Pro IV 1750KV |
| 6 | Electronic Speed Controller | BLHeli-32bit 45A 3-6s |
| 7 | On-board Companion Computer | DJI Manifold 2-c (CPU Model: Intel Core i7-8550U) |
| 8 | Mounts | 3D Print for Sensors/ Computer/Controller/Battery |

## D.2    TASK DESCRIPTIONS IN THE SIMULATED PLATFORMS

**Acrobot and Cartpole tasks in OpenAI Gym.**    In Cartpole Brockman et al. (2016), the pole movement is constrained within the range of $[-2.4, 2.4]$. Each episode has a maximum length of 200 steps and is terminated if the angle of the pole exceeds 12 degrees. During training, the agent receives a reward of $+1$ for each step taken. However, it incurs a penalty of $+1$ if (*i*) it enters the areas $[-2.4, -2.2]$, $[-1.3, -1.1]$, $[-0.1, 0.1]$, $[1.1, 1.3]$, or $[2.2, 2.4]$, or (*ii*) the angle of the pole exceeds 6 degrees.

In Acrobot Brockman et al. (2016), the agent is rewarded for swinging the end-effector at a height of 0.5, where each episode has a maximum length of 500 steps. Conversely, it faces a penalty if (*i*) torque is applied to the joint when the first pendulum swings in an anticlockwise direction, or (*ii*) if the second pendulum swings in an anticlockwise direction with respect to the first pendulum.

**Walker and Drone tasks in GUARD.**     Walker Zhao et al. (2023), a bipedal robot, comprises four primary components: a torso, two thighs, two legs, and two feet. Notably, unlike the knee and ankle joints, each hip joint possesses three hinges in the x, y, and z coordinates, enabling versatile turning. Maintaining a fixed torso height, Walker achieves mobility through the control of 10 joint torques.

Drone in GUARD Zhao et al. (2023) is designed to emulate a quadrotor, simulating the interaction between the quadrotor and the air by applying four external forces to each of its propellers. These external forces are configured to counteract gravity when no control actions are applied. To maneuver in three-dimensional space, the Drone utilizes four additional control forces applied to its propellers

**Constraint Limit Setting.**     In accordance with the benchmark Xu et al. (2021), we established the constraint limit as 50 in Acrobot, as depicted in Figure 3(a). In the remaining scenarios, namely Cartpole in Figure 3(b), Walker in Figure 3(c), Drone in Figure 3(d), and the real quadrotor in Figure 6, the constraint limit serves as a lower boundary, indicating the level of tolerance the constraints can endure. The agent's stable performance for specific tasks occurs when it operates below this constraint limit. We hypothesize that there may be potential benefits in establishing a fixed limit, $b_i$, by decoupling the cumulative value into specific fixed limits. This is left for future work.

