# OpenReview forum: "Constrained Reinforcement Learning as Wasserstein Variational Inference: Formal Methods for Interpretability"
_ICLR.cc/2024/Conference — Submitted to ICLR 2024_

### Official Review · Reviewer_3BVe · 2023-10-21

**Soundness:** 1 poor
**Presentation:** 2 fair
**Contribution:** 2 fair
**Rating:** 3
**Confidence:** 3

**Summary:**

This paper explores the methodology of addressing constrained reinforcement learning by framing sequential decision-making problems as probabilistic inference problems. The focus is on utilizing the Wasserstein distance as a measure to discern the difference between the approximate posterior distribution of the trajectories and the posterior conditioned on the optimality operator. However, due to the computational intractability of the Wasserstein distance, the study employs the Generalized Sliced Wasserstein Distance (GSWD) as a proxy for the original distance. This is subsequently expanded by incorporating a neural network to determine the hyperparameters in GSWD, resulting in a metric named the Adaptive Generalized Sliced Wasserstein Distance (AGSWD).

In the development of the algorithm, the paper introduces a distributional representation to represent the cumulative rewards' distribution. The Distributional Bellman operator is then utilized to upate the critic function within the algorithm. The paper also provides theoretical results for the method, demonstrating that AGSWD is a metric and justifying the global convergence as well as the convergence rate.

Moreover, the paper conducts empirical studies on the performance of the proposed algorithm within both a simulated robotic control environment and a realistic Unmanned Aerial Vehicle control environment. These results highlight a leading convergence rate and robust, safe control performance.

**Strengths:**

1. The overall structure of the paper is generally easy to follow.

2. The paper adeptly integrates a multitude of concepts such as Wasserstein distance approximation, interpretability, distributional Reinforcement Learning (RL), variational inference, constrained RL, and probabilistic inference. These topics have been at the forefront of recent research trends.

3. The empirical results emphasize the superior performance of the proposed Adaptive Wasserstein Variational Optimization (AWaVO) relative to other baseline models in both simulated and realistic environments. This underscores the practical effectiveness of the proposed methodology in real-world applications.

**Weaknesses:**

1. This paper covers a wide variety of concepts. However, there seems to be issues with key definitions and notations, which remain ambiguous or undefined, thus making several key ideas challenging to comprehend. For instance:

- In Formula 2), the subscript 'i' is left undefined, and $b_i$ is also undefined.

-  In Section 4.1, the definition of $q_\theta$ appears flawed. There seems to be a duplicate probability over '$a$'. The second term should likely be $p(s|a,\theta)$ instead.

- In Section 4.1, the definition of $p(\tau|O)$ seems incorrect. The term "equivalence" should likely be replaced with "proportional to" (please refer to formula (1)).

- Formula 3) is incomprehensible without the definitions provided in the appendix. The hyperparameter $l$ and the function $\mathcal(G)$ are undefined in the formula and its introductory explanation. It would be helpful to move this content from the appendix into the main body of the paper.

- In Formula 3), it's unclear what the omitted term in $\cdot$ is. According to the appendix, it's supposed to be a hyperparameter $l$. If that's the case, it's generally inappropriate to omit the hyperparameter in the formula.

- The definition of $R_\mu(l,\tilde{\theta})$ in the appendix is also confusing. Please verify whether the $x$ in the formula should be $\mu$ and provide clarification.

- In Section 4.2, for consistency with the expectation in the objective (4), there should be an expectation in the definition of the trajectory reward $\tilde{r}(\tau)$. Or is it intended to be the random variable of cumulative rewards, with the aim being to estimate the distribution? If so, perhaps it should not be termed a function.

- The inference step of $p(\theta|D)$ in Algorithm 2 needs clarification. Its reference cannot be found in Section 3. Please specify the exact location or formula. Please also distinguish between $\theta$, $\theta^\mu$, and $\theta^Q$. The rationale behind employing different updating methods for $\theta$ and the others also needs to be elucidated.

Typos:
- The "hupersurface" before the formula (3) -> hypersurface.
-  In Section 4.2, the defintion of $\tilde{g}_i(\tau)$ lacks a square bracket on the right.


2. The theoretical results appear to lack key assumptions. Many conclusions are questionable and would benefit from further contemplation.

- The proof of Proposition 1 claims that A-GSWD is a pseudometric rather than a metric. More significantly, it's questionable whether the proof can leverage the fact that "Wasserstein distance is a metric," given that A-GSWD is defined between $\mu$ and $\nu$, not $\mathcal{G}\mu$ and $\mathcal{G}\nu$. If you swap $\mu$ and $\nu$, their corresponding hyperparameters in $\mathcal{G}$ (as output by the neural network) will not be interchanged. Thus, symmetry might not generally hold.

- In the proof of Proposition 3, the expectation following the max operator should be eliminated. If the action is selected based on an existing policy, the max operator becomes irrelevant.

- In Equation (16), the first inequality does not generally hold. It often holds when $\pi_{new}$ corresponds to the argmax Q in the Bellman optimality function. This point needs clarification.

- The Wasserstein distance cannot be related to the discrepancy between quantiles in (18) unless $p$ (parameter) and $d$ (dimension) are both 1 in Wasserstein distance.

- In Formula (19), Lemma 5.1 in Cai et al. (2019) relies on the linearity of the action-value function, which does not generally apply in value networks.

3. The empirical results lack certain key definitions, and their credibility is questionable in several respects:

- The constraint limit in the plots is undefined. Is this intended to represent the tolerance?

- In Figure 3, the plots for cartpole, walker, and guard do not seem to demonstrate signs of convergence. The curves are still rising. An early stop might give the impression that the baseline methods are inferior, but they could potentially surpass the proposed method in the end. The incomplete nature of these experiments means that no definitive conclusions can be drawn.

4. **Motivation.** The implementation of distributional Reinforcement Learning (RL) in this paper lacks a clear rationale. Despite its application, it remains unclear why it's necessary and what benefits it brings to the study.

5. **Novelty.** While the introduction of AGSWD stands as a significant contribution, given that GSWD has already been investigated, simply integrating GSWD with neural network outputs doesn't constitute a major advancement in the field.

**Questions:**

- How should we interpret $g_{rl}$? Its role in Formula (3) isn't clear. Is it intended to be a hyperparameter that defines the plane on which $\Tilde{\theta}$ lies? Or is it a term returned by the RL algorithm?

- In the proof of Proposition 3, what is the relationship between the operators and the new and old policies? Is the output of the operators meant to be the new policy? And, what does $\pi^{\prime}$ represent?

- In Formula (18), what does the symbol $H$ denote?

---

> ### Author Response · Authors · 2023-11-17
> **Response to Reviewer 3BVe (Part 1)**
>
> Thank you for the thoughtful and constructive feedback. Your concerns are addressed below. Please let us know if further clarification is needed.
>
> ***
> >__Weaknesses:__ * In Formula 2), the subscript 'i' is left undefined, and $b_i$ is also undefined.
>
> This aligns with comments from Reviewer x9MY. We have clarified that $b_i$ is a fixed limit for the i-th constraint (see the red highlight in Page 3).
>
> >* In Section 4.1, the definition of $q_\theta$ appears flawed. There seems to be a duplicate probability over '$a$'. The second term should likely be $p(s|a,\theta)$ instead.
>
> We appreciate your thoughtful review. From a macro perspective, the definition of $q_\theta$ is derived from Equation 1, where the probability $p(s,a|\theta)$ originates directly from the second term on the right-hand side. Please consider our response in Reviewer x9MY (Q1) regarding the correctness of the term $p(s,a|\theta)$ in Equation 1 and in underscoring the formulation as a Markov Decision Process.
>
> On a micro level, we do not perceive any redundancy in probability over 'a' as $q(a)$ is not $p(a)$. The individual $q(a)$ serves as a probability function with respect to 'a,' representing the approximation (variational distribution) for the optimality likelihood $p(\tau|\mathcal{O})$.
>
> We appreciate this may have led to some confusion, and to ensure transparency, we incorporate the use of $p(\tau|\theta)$ instead of $p(s,a|\theta)$ in the revised version (see the red highlight in Equation 1, and the definition of $q_\theta$ and $p(\tau|\mathcal{O})$), consistent with the response to Q1 in Reviewer x9MY.
>
> >* In Section 4.1, the definition of $p(\tau|\mathcal{O})$ seems incorrect. The term "equivalence" should likely be replaced with "proportional to" (please refer to formula (1)).
>
> Thank you for your feedback. Indeed, in principle, "proportional to" is a more accurate formulation, and it was duly considered in our initial algorithmic framework construction. However, the implementation posed a challenge. Subsequently, we discovered that the direct "equivalence" rather than "proportional to" yields the same convergence performance in real implementation (as detailed in Lemma 2 and Theorem 2 for theoretical understanding). In response to the reviewer's comment, the revised version use "proportional to" instead of "equivalence" along with a clarification that "equivalence" is used in our actual implementation.
>
> >* Formula 3) is incomprehensible without the definitions provided in the appendix. ...
>
> We move the definitions from the appendix into the main body of the paper. Please refer to the red highlights around Equation 3 for details.
>
> >* In Formula 3), it's unclear what the omitted term in $\cdot$ is. According to the appendix, ...
>
> The symbol $\cdot$ signifies the hyperparameter $l$. We would like to retain the use of the symbol $\cdot$ in Equation 3 for consistency with previous papers, such as [1] (Equation 11) and [2] (Equation 15). This choice is made to clearly depict the slicing process in Wasserstein metrics.
>
> >* The definition of $\mathcal{R}_{\mu}(l,\widetilde{\theta})$ in the appendix is also confusing. Please verify ...
>
> We thoroughly verify that $x$ in the formula is correct as $\mu$ represents probability measures and $x$, defined as $\{x\in \mathbb{R}^d|\langle x,\widetilde{\theta} \rangle=l\}$, denotes hyperplanes. It is essential to integrate over $\mu$ rather than $x$. For further clarification, please refer to both [2] (Equation 6/7) and [3]. We appreciate your observation.
>
> >* In Section 4.2, for consistency with the expectation in the objective (4), there should be an expectation in the definition of the trajectory reward. ...
>
> Yes - we have fixed it in the revised version. Thank you for the comment.
>
> Reference:
>
> [1] Bonneel, Nicolas, et al. "Sliced and radon wasserstein barycenters of measures." Journal of Mathematical Imaging and Vision 51 (2015): 22-45.
>
> [2] Chen, Xiongjie, Yongxin Yang, and Yunpeng Li. "Augmented Sliced Wasserstein Distances." International Conference on Learning Representations. 2021.
>
> [3] Kolouri, Soheil, et al. "Generalized sliced wasserstein distances." Advances in neural information processing systems 32 (2019).

---

> ### Author Response · Authors · 2023-11-17
> **Response to Reviewer 3BVe (Part 2)**
>
> >* The inference step of $p(\theta|D)$ in Algorithm 2 needs clarification. Its reference cannot be found in Section 3. Please specify the exact location or formula. ...
>
> Thanks for your comment.
>
> Regarding the reference of $p(\theta|D)$: this information can be found in the second paragraph of Section 3 (where we state, "the prior probability $p_{D}(\theta)$ is derived from the posterior probability $p(\theta|D)$, where the parameter $\theta$ is inferred from the training dataset $D$"). The term 'inference' here signifies the process of obtaining parameters $\theta$ from the dataset $D$, which, in our specific implementation, corresponds to the training process to determine the parameter values of ${\theta}=[\theta^\mu,\theta^Q]$ in actor and critic networks.
>
> Distinguishing $\theta$, $\theta^\mu$ and $\theta^Q$: The term $\theta$ serves as a general representation of the parameters. Specifically, in our implementation, actor-critic networks are employed, and the parameters are denoted as $\theta^\mu$ and $\theta^Q$ (refer to 'Initialize:' in Algorithm 2).
>
> Rationale of different updating methods: The specific updating methods are elucidated in Section 4.2 (refer to 'Policy Updating'), where the detailed gradients of the actor and critic networks, $\delta_{\theta^\mu}$ and $\delta_{\theta^Q}$, are defined in Equation 10 in Appendix B.
>
> ***
> >* Typos:
>
> We have fixed in the revised version. Thank you!
>
> ***
> __Theoretical results:__
> >* The proof of Proposition 1 claims that A-GSWD is a pseudometric rather than a metric. More significantly, ...
>
> Thanks for this detailed and thoughtful comment. For clarity, it can be proven that A-GSWD is still symmetric, given that it is defined between $\mu$ and $\nu$. To prove symmetry, since the k-Wasserstein distance is a metric [1]:
>
> $$W_k\left(\mathcal{G}\_\mu(\cdot,\widetilde{\theta};g_{rl}),\mathcal{G}\_\nu(\cdot,\widetilde{\theta};g_{rl})\right)=W_k\left(\mathcal{G}\_\nu(\cdot,\widetilde{\theta};g_{rl}),\mathcal{G}\_\mu(\cdot,\widetilde{\theta};g_{rl})\right)$$
>
> Thus, there exists:
>
> $$\bf{A-GSWD}\_{k}(\mu,\nu)=\left(\int\_{\mathcal{R}\_{\widetilde{\theta}}} W\_k\left(\mathcal{G}\_\mu(\cdot,\widetilde{\theta};g_{rl}),\mathcal{G}\_\nu(\cdot,\widetilde{\theta};g_{rl})\right)\mathrm{d}\widetilde{\theta}\right)\^{\frac{1}{k}}=\left(\int\_{\mathcal{R}\_{\widetilde{\theta}}} W\_k\left(\mathcal{G}\_\nu(\cdot,\widetilde{\theta};g_{rl}),\mathcal{G}\_\mu(\cdot,\widetilde{\theta};g_{rl})\right)\mathrm{d}\widetilde{\theta}\right)\^{\frac{1}{k}}=\bf{A-GSWD}\_{k}(\nu,\mu)$$
>
> Therefore, symmetry holds. We emphasize this in the proof of proposition 1 in Appendix C.3 (please see the red highlight in the revised manuscript).
>
> >* In the proof of Proposition 3, the expectation following the max operator should be eliminated. ...
>
> To clarify, $\pi$ represents a policy in the Policy Improvement process (to get the optimized policy that achieves the highest value), not a stationary policy. Consequently, we believe that the expectation should not be eliminated.
>
> >* In Equation (16), the first inequality does not generally hold. It often holds ...
>
> The first inequality in Equation 16 relies on Equation 15 (please note that 'According to Equation 15 and Equation 8, it yields: ...' ). We highlight (red) this relationship in the proof in Appendix C.3. Thank you for highlighting this.
>
> >* The Wasserstein distance cannot be related to the discrepancy between quantiles in (18) unless $p$ (parameter) and $d$ (dimension) are both 1 in Wasserstein distance.
>
> Yes, we agree with this comment. Our proof in Equation 18 is consistent with this statement, and uses the Wasserstein distance based on $W_1$ ($p=1$) and one-dimensional quantiles $q_t^{i,*}$ ($d=1$). We appreciate the detailed feedback, and in the revised version, we highlight (red) this alignment in the proof.
>
> >* In Formula (19), Lemma 5.1 in Cai et al. (2019) relies on the linearity of the action-value function, ...
>
> We agree with this comment. In Equation 19, both the terms $f_i^{(H)}$ and $f_{0,i}^{(H)}$ represent quantiles, serving as local linearization that can be applied with Lemma 5.1 in Cai et al. (2019). In the revised version, we emphasize this point (red highlight) in the proof.
>
> >* The constraint limit in the plots is undefined. Is this intended to represent the tolerance?
>
> The constraint limit relies on the benchmark setting [2] (Acrobot) and the specific tasks, with the intention of facilitating a clear comparison of constraint convergence. We provide one additional sentence in Section 6 to clarify this in the revised version.
>
> Reference:
>
> [1] Villani, Cédric. Optimal transport: old and new. Vol. 338. Berlin: springer, 2009.
>
> [2] Xu, Tengyu, Yingbin Liang, and Guanghui Lan. "Crpo: A new approach for safe reinforcement learning with convergence guarantee." International Conference on Machine Learning. PMLR, 2021.

---

> > ### Author Response · Authors · 2023-11-17
> > **Response to Reviewer 3BVe (Part 3)**
> >
> > >* In Figure 3, the plots for cartpole, walker, and guard do not seem to demonstrate signs of convergence. The curves are still rising. ...
> >
> > We agree with the observation that Figure 3 does not distinctly indicate our proposed AWaVO as the top-performing method. In fact, its performance may not surpass benchmarks, as illustrated in Figure 3(d) where CRPO [1] exhibits better convergence. However, the proposed AWaVO places greater emphasis on two other aspects: i.e., training convergence under uncertainties and decision-making interpretation, as detailed in the second paragraph on Page 8.
> >
> > More significant evidence is given in Figure 6, where we provide comparative experiments for real robot tasks to showcase how AWaVO effectively balances the trade-off between performance and interpretability (i.e., in a more complex sequential decision-making scenario). Please see Figure 4 and discussion on Page 9 for additional details.
> >
> > ***
> > __Motivation.__ The implementation of distributional Reinforcement Learning (RL) in this paper lacks a clear rationale. Despite its application, it remains unclear why it's necessary and what benefits it brings to the study.
> >
> > In principle, distributional representation is an intuitive choice that may offer a more comprehensive and richer value-distribution information, enhancing the stability of the learning process. This is particularly important for Bayesian learning processes, which often face challenges in achieving stable convergence. Additionally, policy optimization tends to be biased toward actions with high-variance value estimates, as some of these values may be overestimated by random chance [2].
> >
> > Moreover, the distributional representation significantly enhances interpretability. As demonstrated in Equation 18 of the proof, it uses quantiles derived from the distributional representation to formally establish the transparency (i.e., quantitive convergence rate) of convergence process outlined in Theorem 2. Ultimately, our experiments, both simulated and real-world, verify a more stable and interpretable learning process, as illustrated in Figure 3 and 6.
> >
> > We appreciate this thoughtful comment, and to provide further clarity we emphasize this in Appendix B.1.
> >
> > ***
> > __Novelty.__ While the introduction of AGSWD stands as a significant contribution, given that GSWD has already been investigated, simply integrating GSWD with neural network outputs doesn't constitute a major advancement in the field.
> >
> > We appreciate the viewpoint and thank the reviewer for recognising the significance of the contribution.
> >
> > ***
> > __Questions:__
> >
> > >* How should we interpret $g\_{rl}$? Its role in Formula (3) isn't clear. Is it intended to be a hyperparameter ...
> >
> > Thanks for the comment. $g_{rl}$ is a measurable mapping $g_{rl}: \mathcal{X}\rightarrow\mathcal{R}_{\widetilde{\theta}}$, where $\mathcal{R}\_{\widetilde{\theta}}$ is a space containing a compact set of all feasible parameters $\widetilde{\theta}$ for $g\_{rl}$ (see Page 4 for details). $g\_{rl}$ can be interpreted as a parameter for the spatial Radon transform $\mathcal{G}$, which is $\mathcal{G}\_{\mu}(l,\widetilde{\theta}) = \int\_{\mathbb{G}^d} \delta(l-\langle g\_{rl},\widetilde{\theta} \rangle) \mathrm{d}\mu$.
> >
> >
> >
> > >* In the proof of Proposition 3, what is the relationship between the operators and the new and old policies? Is the output of the operators meant to be the new policy? And, what does $\pi'$ represent?
> >
> > Yes. The operators $\\{\mathcal{F}\_r,\mathcal{F}\_g \\}$ serve as the link between the optimality likelihood $p(\mathcal{O}|\tau)$ and the accumulated reward & utility function ($\widetilde{r}(\tau)$ & $\widetilde{g}(\tau)$). Essentially, the output of these operators represents the accumulated reward and utility. This can be observed when $Q^{\pi'}(s, a) \geq Q^{\pi}(s, a)$, where $\pi'$ is considered a new policy. In subsequent timestamps, $\pi'$ denotes a policy that satisfies $Q^{\pi'}(s, a) \geq Q^{\pi}(s, a)$.
> >
> > >* In Formula (18), what does the symbol $H$ denote?
> >
> > $H$ is the layers of the neural network, defined in the statement of Theorem 2 (Section 5) and Table 1. We highlight this in red in Equation 18.
> >
> > Reference:
> >
> > [1] Xu, Tengyu, Yingbin Liang, and Guanghui Lan. "Crpo: A new approach for safe reinforcement learning with convergence guarantee." International Conference on Machine Learning. PMLR, 2021.
> >
> > [2] Yecheng Ma, Dinesh Jayaraman, and Osbert Bastani. Conservative offline distributional reinforcement learning. Advances in Neural Information Processing Systems, 34, 2021.

---

> > > ### Comment · Reviewer_3BVe · 2023-11-19
> > >
> > > It appears that most of my concerns are indeed valid, but there are a few places that require further discussion.
> > >
> > > - "The individual q(a) serves as a probability function with respect to 'a,' representing the approximation (variational distribution) for the optimality likelihood ." In fact, I don't think it is correct. Please show me the derivation of your variational inference.
> > >
> > > - Regarding the symmetry of AGSWD, **your new proof appears to reiterate my concern**. Please take note that in your proof, $\mathcal{G}\mu$ and $\mathcal{G}\nu$ should be integral parts of the matrix and they are parameterized by the neural network. The question then is, when you interchange $\mu$ and $\nu$, how can you simultaneously swap $\mathcal{G}\mu$ and $\mathcal{G}\nu$? **Be aware that you metric is evaluating the divergence between $\mu$ and $\nu$, instead of \mathcal{G}\mu(\cdot)$ and $\mathcal{G}\nu$**. Following your line of reasoning, all metrics could be considered symmetric, as long as they can be expressed in some form of the Wasserstein metric.
> > >
> > > - "Consequently, we believe that the expectation should not be eliminated." **Please doublecheck your notation in equation (15)**, if you take the expection over policy, it appears $\mathbb{E}_{\pi}(Q^\pi(s,a))=V^\pi(s)$. then what's the meaning of **$\max_a V^\pi(s)$**? If you want to achieve policy improvement, please just follow the Bellman Optimality Equation.
> > >
> > > - ". As each quantile can be viewed as a form of local linearization". I have not heard of this, please point me to the reference notes or papers.
> > >
> > > - "The constraint limit relies on the benchmark setting [2] (Acrobot) and the specific tasks, with the intention of facilitating a clear comparison of constraint convergence." If "constraint limit" is indeed part of the benchmark, it would be helpful if you could provide more detail about what it signifies or represents, and how you selected these hyperparameters. Initially, I assumed it referred to "tolerance" in the paper, but based on your response, it seems that my assumption was incorrect.
> > >
> > > - Regarding "In Figure 3, the plots for cartpole, walker, and guard do not seem to demonstrate signs of convergence". The proper response is to expand the running episodes of your experiments and present the results once the algorithm under evaluation has converged.

---

> > > > ### Author Response · Authors · 2023-11-21
> > > > **Response to Reviewer 3BVe (Part 1)**
> > > >
> > > > Thank you for the additional questions.
> > > >
> > > > >* "The individual q(a) serves as a probability function with respect to 'a,' representing the approximation (variational distribution) for the optimality likelihood ." In fact, I don't think it is correct. Please show me the derivation of your variational inference.
> > > >
> > > > Let us retrace the derivation of $q(a)$:
> > > >
> > > > 1) In Section 3, leveraging Bayes' theorem, we establish $p(\tau|\mathcal{O})= p(\mathcal{O}|\tau)p(\tau)/p(\mathcal{O})$. By decoupling $p(\tau)$ into $p(\tau|\theta)p(\theta|D)$ (and $p(\mathcal{O})$ is a constant, $\mathcal{O}\_t \in\\{0,1\\}$), where $\theta$ and $D$ represent the parameters and the observed dataset sampled from sensors, respectively, we arrive at $p(\tau|\mathcal{O})\propto p(\mathcal{O}|\tau)p(\tau|\theta)p_D(\theta)$, i.e., Equation 1, with $p_D(\theta)$ denoting $p(\theta|D)$.
> > > >
> > > > 2) Moving on to Section 4.1, we introduce variational inference: $D(q_\theta(\tau)||p(\tau|\mathcal{O}))$. Based on Equation 1, we __construct__ a formulation of the variational distribution $q_\theta(\tau)=q(a)p(\tau|\theta)p_D(\theta)$ by following the structure of Equation 1. Here, $q(a)$ approximates the optimality likelihood $p(\mathcal{O}|\tau)$.
> > > >
> > > > 3) Importantly, in the latter part of Section 4.1, we extend the Wasserstein distance $A-GSWD$ (integrating GSWD with PPO-DR) to the inference problem as ${\bf{GSWD}}(q_\theta(\tau), p(\tau|\mathcal{O}))$.  In Appendix C.1, we give the derivation of how we transform the original minimization problem $\arg \underset{q\_\theta(\tau)}{\rm{min}} {\bf{GSWD}}\_{k}\left(q\_\theta\left(\tau\right),p\left(\tau|\mathcal{O}\right)\right)$ (between the two posteriors) into the new minimization problem $\arg \underset{q_\theta(\tau)}{\rm{min}}{\bf{GSWD}}_{k}\left(q\left(a\right),p\left(\mathcal{O}|\tau\right)\right)$ (between the optimality likelihood $p(\mathcal{O}|\tau)$ and its approximation $q(a)$). Thus, our approach involves solving the inference problem by determining the approximation $q(a)$, as illustrated in Figure 4. This is why we state in the previous response: "The individual q(a) ... ... representing the approximation (variational distribution) for the optimality likelihood."
> > > >
> > > > >* Regarding the symmetry of AGSWD, ... ..., they can be expressed in some form of the Wasserstein metric.
> > > >
> > > > The proposed A-GSWD is built on Sliced Wasserstein Distances (SWD) [1], wherein both symmetric (as outlined in Section 2.3 in [1]) and its variants Generalized Sliced Wasserstein Distances (GSWD) (as discussed in Remark 1 in [2]) are established. In A-GSWD, we extend the linear hyperplane to a non-linear hypersurface. Additionally, Augmented Sliced Wasserstein Distances (ASWD) [3] provide a similar proof of symmetry, specifically defined on the hypersurface as well (refer to the proof of Theorem 1 in [3]).  Consequently, we believe the proposed A-GSWD is symmetric, and the variant metrics could be considered symmetric as long as their forms inherit the structure of SWD, as exemplified in GSWD [2] and ASWD [3].
> > > >
> > > > >* ... ..., Please doublecheck your notation in equation (15), ... ... follow the Bellman Optimality Equation.
> > > >
> > > > Referring to Section 2.1 in Bellemare 2017 [4], where, according to the Bellman optimality equation (Eq. 3 in [4]), we prefer to similarly keep the expectation but remove its subscript $\pi$, i.e., using $\mathbb{E}$ instead of $\mathbb{E}\_\pi$ in our Eq. 15.
> > > >
> > > > >* ". As each quantile can be viewed as a form of local linearization". ... ...
> > > >
> > > > Certainly. We state "each quantile can be viewed as a form of local linearization" because of the nature of quantiles.
> > > >
> > > > Regression quantile was introduced by Koenker and Bassett (1978) [5] as a means of estimating conditional quantiles in linear regression models. Then, a local linear estimator of the conditional quantile was proposed in [6], which indicates the quantile can be viewed as a form of local linearization. Additionally, the theory of quantile regression is explained in [7] (cf Subsection 1.4 'Preview of Quantile Regression' (p 10), wrt "regression quantiles", where the solutions have properties that “follow immediately from well-known properties of solutions of linear programs".

---

> ### Author Response · Authors · 2023-11-21
> **Response to Reviewer 3BVe (Part 2)**
>
> >* "The constraint limit relies on the benchmark setting (Acrobot) and the specific tasks, ... ...
>
> Thank you for this comment. Initially, we established the constraint limit with the intention of facilitating a straightforward comparison of constraint convergence, as the constraint limit is introduced in the benchmark. After reviewing your comment, we recognize a potential connection with "tolerance", wherein the agent's stable performance for specific tasks occurs when it operates below the constraint limit. In our view, the constraint limit serves as a lower boundary that indicates the level of tolerance the constraints can withstand. We believe there might be advantages in setting a fixed limit, $b_i$, by decoupling the cumulated value into each specific fixed limit. This aspect will be explored in future work. We add this clarification in Section 6 and Appendix D.2.
>
> >* Regarding "In Figure 3, the plots for cartpole, walker, and guard do not seem to demonstrate signs of convergence". The proper response is to expand the running episodes of your experiments and present the results once the algorithm under evaluation has converged.
>
> Thank you for this suggestion. We have expanded our experiments in cartpole, walker and drone to showcase the convergent results. We have modified Figure 3 in the updated version to reflect this.
>
> Reference:
>
> [1] Kolouri, Soheil, Gustavo K. Rohde, and Heiko Hoffmann. "Sliced wasserstein distance for learning gaussian mixture models." Proceedings of the IEEE Conference on Computer Vision and Pattern Recognition. 2018.
>
> [2] Kolouri, Soheil, et al. "Generalized sliced wasserstein distances." Advances in neural information processing systems 32 (2019).
>
> [3] Chen, Xiongjie, Yongxin Yang, and Yunpeng Li. "Augmented Sliced Wasserstein Distances." International Conference on Learning Representations. 2021.
>
> [4] Bellemare, Marc G., Will Dabney, and Rémi Munos. "A distributional perspective on reinforcement learning." International conference on machine learning. PMLR, 2017.
>
> [5] Koenker, Roger, and Gilbert Bassett Jr. "Regression quantiles." Econometrica: journal of the Econometric Society (1978): 33-50.
>
> [6] Gannoun, Ali, Jérôme Saracco, and Keming Yu. "Comparison of kernel estimators of conditional distribution function and quantile regression under censoring." Statistical Modelling 7.4 (2007): 329-344.
>
> [7] Koenker, Roger. Quantile regression. Vol. 38. Cambridge university press, 2005.

---

### Official Review · Reviewer_e4UV · 2023-10-31

**Soundness:** 3 good
**Presentation:** 2 fair
**Contribution:** 3 good
**Rating:** 6
**Confidence:** 2

**Summary:**

Treating RL problems in terms of variational inference is a promising approach
to building interpretable policies, but it suffers from high computational
expense. In particular, the cost of computing distance metrics between
probability distributions over high dimensional spaces is intractable. This
paper proposes a new metric (A-GWSD) on probability distributions which can be
computed efficiently and used to solve RL-as-inference problems. This metric is
used to solve constrained RL problems. This algorithm is shown to theoretically
satisfy a global convergence property, and empirically achieves similar rewards
and constraint violation rates to state-of-the-art safe RL algorithms. In
addition, the proposed approach yields a clear interpretation of policy actions
based on conditional probability evaluation.

**Strengths:**

Interpretability is a key challenge in reinforcement learning, and this paper
offers a promising approach to providing explanations for agent behavior.

Constraints are also important in real-world reinforcement learning. This paper
proposes a way to extend a popular RL framework (RL as inference) to the
constrained setting.

The experimental results show that improvements in interpretability can be
achieved with minimal sacrifices in terms of rewards or constraint violations.

**Weaknesses:**

Given that the main improvement over prior work is in intpretability, I would
have liked to see more discussion of interpretability in the experimental
results. There is some information in Figure 4 (but see the questions below),
but I think the paper would benefit from some discussion of intepretability in
the other environments.

**Questions:**

I found it difficult to understand Figure 4, which is where the key advantage of
this approach over prior work is shown. Some explanation of why this diagram is
evidence of improved interpretability would be helpful. I can see that roughly
speaking, when the wind estimation in 4(a) is higher, the probability in 4(b) is
higher as well, which demonstrates some sensitivity to the wind. But the two
graphs do not line up that closely and I'm not sure if I'm interpreting them
correctly. What should I be looking for in this figure?

---

> ### Author Response · Authors · 2023-11-17
> **Response to Reviewer e4UV**
>
> We appreciate your positive feedback. Your concerns are addressed below. Please let us know if further clarification is needed.
>
>
>
> ***
>
> >* __Question__: I found it difficult to understand Figure 4, which is where the key advantage of this approach ... ... What should I be looking for in this figure?
>
>
>
> Figure 4 (a) is provided as a reference; i.e., the curves give our estimated values of external aerodynamic forces (winds) in real time. These are derived from the signals collected from onboard sensors. In Figure 4 (b), we illustrate the quantitative impact of external forces, denoted as $L\_0$, on the current sequential decisions, i.e., the planned trajectory, denoted as $\tau$, and the corresponding Pulse Width Modulation signals that are fed to the motors.
>
>
>
> Figure 4 (b) becomes particularly important when the agent makes suboptimal decisions leading to events like crashes or collisions in quadrotors. These curves can be leveraged for interpretation and conducting quantitative analyses of specific environmental factors, such as winds and obstacles, in order to understand their magnitudes of influence on current decisions taken.
>
>
>
> In more detail, consider Reference State 02 (RS 02) as an example. In Flight Task 3 (FT 3), represented by the red curve, we observe how the external forces (i.e., the aerodynamic effects generated by the winds and obstacles) influence the current trajectory planning decisions. With RS 02 situated in an area featuring a combination of wind and obstacles, the value is approximately $0.5$. This implies that the current planned trajectory (sequential decisions) has about a $50%$ probability of being influenced by the aerodynamic effects. Comparing FT 1 and FT 2, where the values in RS 02 are approximately $0.24$ (FT 1) and $0.15$ (FT 2), we can decouple the aerodynamic effects generated by the wind on the body (FT 1) and obstacles (FT 2), respectively. Quantitatively, the red $p(\tau|L_0)\_{FT3}$ is approximately equal to the sum of $p(\tau|L_0)\_{FT1}$ (only wind) and $p(\tau|L_0)\_{FT2}$ (only obstacles).
>
>
>
> ***
>
> >* __Weaknesses:__ Given that the main improvement over prior work is in intpretability, ... There is some information in Figure 4 (but see the questions below), but I think the paper would benefit from some discussion of intepretability in the other environments.
>
>
>
> Thank you for raising this. Although there is limited scope to expand in this paper, we are excited to implement and examine our work for various real-world environments and applications, particularly focusing on safety-critical scenarios with variable uncertainties. Examples of such applications include self-driving ground vehicles, aerial delivery aircraft systems and other robotic platforms. We have commented on this as part of our future work in Section 7.

---

> > ### Comment · Reviewer_e4UV · 2023-11-20
> >
> > Thank you for the response. This helps clarify the interpretability claims for me. I look forward to seeing future work in more diverse and safety-critical experimental settings.
> >
> > Am I correct in interpreting FT 3 as a kind of combination of FT 1 and FT 2 (in that FT 1 has wind and FT 2 has obstacles)? If so, is there a reason I shouldn't expect $p(\tau | L_0)$ for FT 3 to _always_ be approximately the sum of $p(\tau | L_0)$ for FT's 1 and 2, rather than just at RS 02? It seems this property does no hold at other points along the trajectory.

---

> > > ### Author Response · Authors · 2023-11-21
> > > **Response to Reviewer e4UV**
> > >
> > > Thank you for the encouraging comments and additional questions.
> > >
> > > >* Am I correct in interpreting FT 3 as a kind of combination of FT 1 and FT 2 (in that FT 1 has wind and FT 2 has obstacles)? ... ... does no hold at other points along the trajectory.
> > >
> > > Yes. The specific Reference State (RS) in Flight Task (FT) 1-3 is not, however, to be consistently interpreted as a combination. Consider the following two aspects:
> > >
> > > 1) The real-time trajectory $\tau$ (i.e., sequential decisions) varies across distinct flight tasks, as illustrated in the specific planned trajectory in Figure 4 (top). In this context, $\tau$ is dynamically planned to accomplish objectives such as obstacle avoidance and minimize energy consumption.
> > >
> > > 2) The environmental factor, specifically denoted as $L_0$, which represents the aerodynamic effect, does not always conform to the combination FT 1 + FT 2 = FT 3. This is evident in instances like RS 03 & 04, where the flight environment surrounding the quadrotor remains similar.
> > >
> > > We can consider RS 02 to illustrate this point, as it represents a scenario where the two aspects are approximately satisfied. In RS 02, the trajectory around the quadrotor follows a similar pattern, and the surroundings of the quadrotor can be quantitatively interpreted as an approximate combination.

---

> > > > ### Comment · Reviewer_e4UV · 2023-11-21
> > > >
> > > > Thank you for the clarification. I will keep my score in favor of acceptance. In general, I like this technique and the results presented are promising, but I still believe the paper would be stronger with a more extensive discussion of interpretability.

---

### Official Review · Reviewer_x9MY · 2023-11-02

**Soundness:** 3 good
**Presentation:** 2 fair
**Contribution:** 3 good
**Rating:** 6
**Confidence:** 3

**Summary:**

The authors propose a distributional approach to infer a policy in the constrained reinforcement learning setting. The central idea is to alternatively perform Wasserstein variational inference (WVI) and distributional policy optimization (PPO-DR). WVI learns a variational approximation $q(a)$ to the optimality likelihood $p(O|\tau)$ (similar to the control as inference framework [1]) while PPO-DR maximizes the expected reward within the feasible region, and minimizes the expected constraint outside the feasible region (using distributional networks). The variational approximation $q(a)$ is represented by the critic distribution, while the Wasserstein distance is computed using the hypersurfaces given by the actor distribution. Overall, the idea of using adaptive GSWD for distributional constrained RL is interesting, and supported by theoretical and empirical results, but the work can be improved in terms of clarity.

**Strengths:**

1. Extensive discussion on convergence results
2. The variational inference process is interesting, and it minimizes a Wasserstein distance (better for distributional RL) while ensuring that the actor distribution is used in computing the distance.
3. Real world experiments

**Weaknesses:**

On page 3, last line: "where, upon satisfying the constraints, the agent enters a state considered as safe". In my understanding, in the expected constraint formulation (Equation 2), if the constraints are satisfied, then the policy is considered "safe". "Safe/unsafe" states typically refer to a CMDP formulation with constraint sets, i.e. a part of the state-action space is safe, and the rest is unsafe. While it is useful to obtain a graphical model framework equivalent to the control as inference framework [1], a more appropriate justification of Figure 1 should have been that the optimality variables are influenced by the constraint, rather than saying that the agent enters a safe state. This means that $p(\tau|O)$ becomes 0 as soon as the constraint is not satisfied in expectation. The authors do not model the probability in this way, but rather learn a policy by maximizing reward when within the feasible region, and minimizing constraint when outside the feasible region (to get within the feasible region). This makes sense intuitively, but equivalence to constrained RL (equation 2) is not formally established.

**Questions:**

1. In Equation 1, isn't it more appropriate to use $p(\tau|\theta)$ instead of $p(s,a|\theta)$ for the second term of the right hand side?
2. In equation 2, what is $b_i$? Maybe I missed it, but why are there two thresholds?
3. (Suggestion) Page 4, after equation 3: "$\tilde \theta$ are the returns from the actor networks" could be re-worded. Returns have a specific meaning in RL literature, and in this context, I think return just means the output of the actor network, and not the usual return. (please ignore if my understanding is incorrect)
4. Maybe I missed this, but how are the individual $p(L_i|D)$ obtained (in the flight task setup)?

**References**

1. Reinforcement learning and control as probabilistic inference: Tutorial and review, Levine (2018)

**Details Of Ethics Concerns:**

_

---

> ### Author Response · Authors · 2023-11-17
> **Response to Reviewer x9MY**
>
> Thank you for the thoughtful and positive feedback. Your concerns are addressed below. Please let us know if further clarification is needed.
>
> ***
> >* __Weaknesses:__
>
> We are grateful for your assessment of our research. We agree with many of your statements like '"Safe/unsafe" states typically refer to a Constrained Markov Decision Process formulation with constraint sets' and 'learn a policy by maximizing reward when within the feasible region, and minimizing constraint when outside the feasible region'. However, we would like to address some specific statements to improve our shared understanding:
>
> > 1. a more appropriate justification of Figure 1 should have been that the optimality variables are influenced by the constraint,
>
> Yes - we concur that 'the optimality variables are influenced by the constraint'. To elaborate, when $\mathcal{O}=1$, it signifies that the trajectory $\tau$ is __not only optimized but also satisfied the constraints__. In other words, it denotes an ideally optimized trajectory that meets the prescribed constraints. A clear explanation has been highlighted (red) at the outset of Section 3 on page 3. Thank you for highlighting this.
>
> > 2. rather than saying that the agent enters a safe state. ... This makes sense intuitively, but equivalence to constrained RL (equation 2) is not formally established.
>
> It is important to clarify that, in principle, constrained Reinforcement Learning (RL) cannot guarantee full adherence to constraints. Throughout the learning process, an agent is awarded or penalized, pushing it towards a 'safe' state. Consequently, there exists a certain probability of violating constraints in states or actions. This challenge poses a significant barrier for applying constrained RL in safety-critical applications.
>
> More importantly, while AWaVO does not ensure absolute safety, we offer a clear theoretical insight in Theorem 2. This theorem quantitatively interprets the violation: the convergence rate will follow a sublinear $\Theta(1/\sqrt{T})$ if constraints are violated with an error of $\Theta(1/{m^{\frac{H}{4}}})$ (which decreases as the width and layer count of the neural network, denoted as $m$ and $H$ respectively, increase), with a probability of at least $1-\delta$. For detailed information, we refer to Equations 26-28 in the proof of Theorem 2.
>
> Consequently, we believe that AWaVO is not 'equivalent' to Equation 2, but surpasses constrained RL in terms of interpretability.
>
> ***
> >* __Questions:__ 1. In Equation 1, isn't it more appropriate to use $p(\tau|\theta)$ instead of $p(s, a|\theta)$ for the second term of the right hand side?
>
> Yes - we appreciate this insightful interpretation of Equation 1. Using $p(\tau|\theta)$ in the second term would bring Equation 1 closer in form to Bayes' theorem, making it more intuitively understandable. Essentially, the formation process of the second term outlines a Markov Decision Process, wherein $p(a_t|s_t,\theta)$ and $p({s}_{t+1}|{s}_t,{a}_t)$ represent the policy state $\pi$ and transitions, respectively.
>
> The rationale behind our use of $p(s, a|\theta)$ in this context lies in the Markov property, indicating that the present state encompasses all the information necessary to predict future states, irrespective of historical states. Consequently, we adopt $p(\tau|\theta)$ in Equation 1 (see the red highlight in our revision) and explicitly clarify the Markov property for better comprehension.
>
> >2. In equation 2, what is $b_i$? Maybe I missed it, but why are there two thresholds?
>
> $b_i$ is a fixed limit for the i-th constraint, while $\tau_c$ is the tolerance. The reason we use the tolerance $\tau_c$ here is for the convergence process when the approximated constraints $J_{g,i}(\pi)$ violate the fixed constraint limit $b_i$, necessitating the bounding of errors (cf. Equation 27 in the proof of Theorem 2).
>
> >3. (Suggestion) Page 4, after equation 3: ... Returns have a specific meaning in RL literature, ... I think return just means the output of the actor network, ...
>
> Thanks for highlighting this. We use 'outputs' in the revised paper when conveying this concept.
>
> >4. Maybe I missed this, but how are the individual $p(L_i|D)$ obtained (in the flight task setup)?
>
> The individual $p(L_i|D)$ represents the agent's capability to detect the factor $L_i$ within the environments $D$, typically acquired from sensors or perception modules. In the context of our flight task, $L_0$ represents external forces, and $D$ represents depth images sampled from a depth camera, namely the Intel Realsense D435i. Therefore, $p(L_i|D)$ individually represents the estimation of external forces $L_i$ within the environment $D,' and this estimation is obtained through VID-Fusion [1].
>
> Reference:
>
> [1] Ding, Ziming, et al. "Vid-fusion: Robust visual-inertial-dynamics odometry for accurate external force estimation." 2021 IEEE International Conference on Robotics and Automation (ICRA). IEEE, 2021.

---

> > ### Comment · Reviewer_x9MY · 2023-11-22
> > **Rebuttal response**
> >
> > In the new text, the authors clarify that "$\mathcal O_t=1$ signifies that the trajectory $\tau$ is both optimized and compliant with the constraints". While this tries to address my earlier comment that optimality variables are influenced by the constraint, I don't think this is still the right formulation. With no constraints, the optimality variables induce a distribution $p(\mathcal O_t=1|s_t,a_t)=e^{r(s_t,a_t)}$ as given in [1], but with expected constraints, how would this change? I suspect that since expected constraints are not satisfied per step, we cannot write a similar expression for each $\mathcal O_t$.
> >
> > >  Throughout the learning process, an agent is awarded or penalized, pushing it towards a 'safe' state.
> >
> > Again, since this is the expected constraint formulation, there is no safe state. Safeness is ensured if the current policy yields trajectories that in expectation (across several trajectories) satisfies the expected constraint. I will still suggest to change this wording, if possible.
> >
> > Thanks for the rest of the clarifications. I'm inclined to keep my score as some of my concerns remain.
> >
> > **References**
> >
> > 1. Reinforcement learning and control as probabilistic inference: Tutorial and review, Levine (2018)

---

> ### Author Response · Authors · 2023-11-22
> **Response to Reviewer x9MY - thank you for your response**
>
> We value your additional feedback, and the remaining concerns you raised are addressed below. Please let us know if further clarification is needed.
>
> >* With no constraints, the optimality variables induce a distribution ... ... how would this change?
>
> Thank you for your thoughtful comment. To enhance clarity, we have revised the statement in Section 3 (page 3), introducing the optimality family $\mathcal{O}\_{t}=\\{\mathcal{O}\_{r,t},\mathcal{O}\_{g,t}\\}\in\\{0,1\\}$ and specifying that $\mathcal{O}\_{r,t}=1$ and $\mathcal{O}\_{g,t}=1$ signify the trajectory $\tau$ is optimized and compliant with the constraints, respectively. Then, we present the following clarifications:
>
> 1) With no constraints, the optimality likelihood induce a distribution as $p(\mathcal{O}\_{r,t}=1|s\_t,a\_t)$;
>
> 2) With expected constraints, the optimality likelihood induce a distribution as $p(\mathcal{O}\_{g,t}=1|s\_t,a\_t)$.
>
> Please also refer to Equation 4 and Section 4.2 for a slightly revision.
>
> >* ... ... Safeness is ensured if the current policy yields trajectories that in expectation (across several trajectories) satisfies the expected constraint. I will still suggest to change this wording, if possible.
>
> Certainly, and thank you for this suggestion. To provide clarity on the formulation of expected constraints, we have revised the statement in Section 3, found at the beginning of page 4.

---

> ### Comment · Reviewer_x9MY · 2023-11-23
> **Further comments regarding response**
>
> Thanks for your clarification. I think the language is better now, but again, like I mentioned before, the constraints are expected, so they hold for a batch of trajectories, not for a single trajectory. Therefore, the optimality distributions eg. $p(\mathcal O_g|\tau)$ should rather be for multiple trajectories, not a single trajectory, right?

---

> ### Author Response · Authors · 2023-11-23
> **Response to Reviewer x9MY - thank you for your addtional comment**
>
> Thanks for your comment. Indeed, both the distributions $p(\mathcal{o}\_r|\tau)$ and $p(\mathcal{o}\_g|\tau)$ hold across multiple trajectories. We have made a slight revision in Section 3, located in page 3, to clarify the 'expected formulation' within the context of 'several trajectories'. If further clarification is needed please let us know.

---

### Author Response · Authors · 2023-11-17
**Global Response by Authors: New Results & Paper Updates**

We thank all the reviewers for their thoughtful, detailed and constructive feedback. In the responses below, we address each raised concern individually. A revised version has been uploaded, and variations from the original submission are highlighted in red.

---

### Meta-Review · Area_Chair_BKQn · 2023-12-06

**Metareview:**

The submitted paper proposes Adaptive Wasserstein Variational Optimization (AWaVO), an approach for achieving interpretability of RL by considering RL as probabilistic inference and connecting to formal methods. In particular, claims regarding the interpretability of guaranteed convergence, training transparency, and sequential decisions are made.

Clearly, enabling interpretability of RL is a worthwhile and important goal, and the paper connects many different concepts (Wasserstein distance approximation, interpretability, distributional RL, variational inference, constrained RL, and probabilistic inference) to achieve it, demonstrating improved interpretability while maintaining high performance in an empirical evaluation. However, there were certain issues regarding the theoretical derivation/proofs the formulation of constrained RL as probabilistic inference that could not be resolved during rebuttal. Thus I am recommending the rejection of the paper and encourage the authors to improve their paper in line with the reviewers' comments, paying particular attention to carefully addressing and better explaining their theoretical derivations.

**Justification For Why Not Higher Score:**

We should not accept papers with incorrect or unclear theoretical statements.

**Justification For Why Not Lower Score:**

N/A

---

### Decision · Program_Chairs · 2024-01-16

Reject